# Growth history leaves a geometric trace in puzzle cells

Nicola Trozzi [1,2,3], Brendan Lane [2], Alice Perruchoud[1], Frances Clark[4], Lukas Hoermayer[1], Andrea Meraviglia [1], Tammo Reichgelt [5], Adrienne H K Roeder [4], Dorota Kwiatkowska [6], Adam Runions [7], Richard S Smith [2,3✉] & Mateusz Majda [1✉]

## Abstract

Puzzle-shaped epidermal cells can reduce mechanical stress during organ growth and, as shown here, can also record tissue expansion history in their outlines. By combining mechanical simulations with time-lapse imaging, we find that transitions from directional to isotropic expansion induce new lobes along the previous growth axis, and that reversing the sequence of anisotropic and isotropic phases yields hybrid shapes that preserve the sequence of growth phases. In maize, model predictions closely match live imaging, and in *Arabidopsis*, final lobe patterns correlate more strongly with growth history than with cell size alone. Genetic or pharmacological perturbations that reduce lobing are associated with constrained leaf expansion or compensatory elongation, consistent with a mechanical role. A broad survey of living and fossil vascular plants indicates that the capacity to form puzzle-shaped cells is widespread and developmentally plastic, suggesting that single snapshots of leaves can reflect prior growth dynamics across species. Together, these findings show that puzzle cells transform cell geometry into a living record of how tissues grow.

**Keywords** Cell Shape; Growth; Mechanical Stress; Pavement Cells; Puzzle Cells
**Subject Categories** Cell Adhesion, Polarity & Cytoskeleton; Development; Plant Biology

## Introduction

The shapes of plant organs, whether broad or elongated, are often reflected in the shapes of individual cells, suggesting a close relationship between organ growth patterns and cellular morphology (Hu et al, 2024; Hülskamp et al, 1998; Zuch et al, 2022). Studies using *Arabidopsis thaliana* mutants have demonstrated that changes in organ shape are accompanied by corresponding changes in cell shape, and vice versa, highlighting the interdependence of growth dynamics across scales (Hake et al, 2004; Qiu et al, 2002; Tasker-Brown et al, 2024; Vlad et al, 2014). For simple cell geometries, this relationship is intuitive; elongated organs tend to have long, thin cells. However, many epidermal tissues form puzzle-shaped cells, and the link between organ shape and cell shape can appear, at first glance, to be less direct. Integrating experimental biology with theoretical modeling has advanced our understanding of how genes, mechanics, and growth interact, and it emphasizes the need to view tissues and organs as integrated systems rather than a collection of isolated cells. Although molecular regulation acts at the cellular level, the specified growth of individual cells must conform to neighboring cells, and the resulting growth can differ because of mechanical conflicts (Rebocho et al, 2017). The same principle applies to cell shape, which emerges from collective tissue dynamics rather than solely from the molecular regulation of individual cells (Majda et al, 2021; Sapala et al, 2018; Vőfély et al, 2019).

Endoreduplication, characterized by successive rounds of DNA replication without cell division, results in substantial cell enlargement, increasing the mechanical demands on cell walls to withstand higher turgor-driven stress in larger cells (Edgar et al, 2014; Roeder et al, 2010). In stems and hypocotyls, cells typically elongate along the main growth axis (Baskin, 2005; Cosgrove, 2022; Gendreau et al, 1997). The elongated geometry naturally limits stress and can be approximated by the largest empty circle (LEC), defined as the largest circle that fits within a cell outline. Although the LEC is computed from 2D outlines, 3D finite element modeling showed that it provides a good proxy for turgor-induced stress by capturing the maximal unsupported span of the outer periclinal wall while integrating surface curvature and 3D shape effects (Sapala et al, 2018). In the leaves analyzed here, organ scale curvature differs only modestly between the adaxial and abaxial surfaces at the stage when lobing begins, and that curvature is small compared with the strong cell scale curvature generated by lobes and necks, so the LEC computed from 2D outlines remains a robust proxy for maximal unsupported span under turgor. In contrast, nearly isotropic growth in broad aerial organs like leaves presents a distinct challenge, because isotropic expansion can generate large

[1]The Mechanobiology Laboratory, Department of Plant Molecular Biology, University of Lausanne, CH-1015 Lausanne, Switzerland. [2]Department of Computational and Systems Biology, John Innes Centre, Norwich NR4 7UH, UK. [3]School of Biological Sciences, University of East Anglia, Norwich NR4 7TJ, UK. [4]Weill Institute for Cell and Molecular Biology and School of Integrative Plant Science, Section of Plant Biology, Cornell University, Ithaca, NY 14853, USA. [5]Department of Earth Sciences, University of Connecticut, Beach Hall, 354 Mansfield Rd, Storrs, CT 06269, USA. [6]Institute of Biology, Biotechnology and Environmental Protection, Faculty of Natural Sciences, University of Silesia, 40-032 Katowice, Poland. [7]Department of Computer Science, University of Calgary, Calgary, AB T2N 1N4, Canada. ✉E-mail: Richard.Smith@jic.ac.uk; Mateusz.Majda@unil.ch

unsupported regions that elevate wall stress and increase rupture risk (Hamant and Traas, 2010; Kierzkowski and Routier-Kierzkowska, 2019; Malivert et al, 2021). Epidermal cells often address this by adopting a puzzle-shaped morphology with interlocking lobes. Lobes decrease the LEC, reduce mechanical stress, and help maintain tissue integrity under tension (Bidhendi et al, 2019; Bidhendi and Geitmann, 2019; Bidhendi et al, 2023; Majda et al, 2017; Sapala et al, 2018). 3D stress analyses using finite element modeling support a functional role for these shapes, showing that lobed cells experience reduced wall stress during isotropic expansion. Together, these results suggest that puzzle shape formation can act as a mechanical adaptation to accommodate endoreduplication-driven cell enlargement during organ growth (Sapala et al, 2018).

At the cellular level, plant cells are tightly connected by their cell walls, meaning that their growth must be coordinated to maintain tissue integrity. This coordination is influenced by the interplay between internal turgor pressure and the mechanical properties of the cell wall, which together constrain how neighboring cells can grow relative to each other. Turgor pressure generates tensile in-plane stress across the cell wall, which resists expansion while the wall is remodeled in a controlled fashion. This remodeling includes the regulated sliding of cellulose fibers at specific attachment points, allowing the cell to grow without compromising its structure (Coen and Cosgrove, 2023; Cosgrove, 2022; Zhang et al, 2021). Cortical microtubules guide the localized deposition of cellulose microfibrils, thereby stiffening specific regions of the cell wall and reinforcing cell structure. In puzzle-shaped cells, microtubules accumulate in the narrow neck regions between protruding lobes, restricting local expansion and shaping the lobed contour, whereas in elongated cells microtubules align transversely to limit lateral growth (Baskin, 2005; Panteris and Galatis, 2005; Paredez et al, 2006; Sampathkumar et al, 2014). One hypothesis tested in a spring-based model was that localized growth restrictions create indentations that then attract microtubules and become reinforced, establishing a feedback loop. Due to growth conflicts, neighbor cells will have lobes that correspond to the indentations, with coordination emerging as a mechanical effect (Sapala et al, 2018). Computational simulations based on these ideas have reproduced a variety of cell shapes and the significant variation found in nature, demonstrating their effectiveness in modeling the dynamics of puzzle cell shape acquisition. Although unintuitive, the model shows that anisotropic growth leads to long, thin cells without lobes, whereas more isotropic growth yields complex puzzle-shaped cells.

If puzzle cell shape is an adaptation to a mechanical constraint based on stress, then one prediction would be that most plants would possess the ability to make puzzle cell shapes. A positive correlation between cell area and lobe number has been observed in *Arabidopsis* and several other species (Sapala et al, 2018). However, in a broader survey of vascular plants (Vőfély et al, 2019), no correlation between cell area and cell lobing was found, and the average lobeyness remained low even in species with large cells, indicating that highly lobed cells, like those in *Arabidopsis* leaves or cotyledons, are relatively rare. They proposed that epidermal cell shapes do not have a conserved function across vascular plants, but rather may serve diverse functional roles across taxa, representing species-specific adaptive strategies shaped by different ecological and evolutionary pressures.

In this work, we investigate whether the ability to form puzzle-shaped cells is broadly conserved across vascular plants, and how developmental stage, organ identity, and environmental conditions influence their occurrence. By combining modeling simulations, live imaging, and a large-scale comparative survey, we examine how growth history and mechanical constraints interact to shape epidermal cell morphology, and whether cell shape retains a signature of past growth.

## Results

### Impact of organ growth dynamics on puzzle cell formation

The spring model proposed by Sapala et al (2018) provides a mechanistic explanation for the formation of shapes that maintain small LECs, ranging from puzzle-shaped to elongated cells. This model offers a framework not only for understanding how shape emerges at the cellular level but also provides an unexplored opportunity to study the interaction of mechanical constraints and growth dynamics at the organ scale. To empirically validate this model, we examined the distribution of cortical microtubule orientation in experimentally observed epidermal pavement cells. By analyzing the cell wall morphology and microtubule patterns (Fig. 1A) and using these observations to create segmented cell models for simulation, we replicated the microtubule dynamics predicted by the model. Simulated connections, which correspond to cortical microtubules and downstream cellulose deposition, were more concentrated in the indentations and notably sparse in the lobed regions of the cells (Fig. 1A,B). These results demonstrate that the Sapala et al (2018) model can generate realistic puzzle cell shapes with microtubule patterns consistent with experimental observations.

Growth anisotropy significantly influences epidermal cell shape, generally producing elongated cells under anisotropic growth and puzzle-shaped cells under isotropic growth (Sapala et al, 2018). Building upon this finding, we further explored how the sequence of anisotropic and isotropic growth phases affects cell morphology, even when the total cumulative growth is the same. In these simulations, the growth schedule was predefined and applied uniformly across the tissue. We conducted computational simulations starting from an initial template of isodiametric cell shapes (Fig. 1C), expanding the cells 16 times vertically and ten times horizontally (Appendix Fig. S1). In the first scenario, we applied uniform anisotropic growth rates to the tissue with a ratio of 1.6:1. This resulted in slightly elongated cells with lobes similar to the isotropic case (Fig. 1D; Movie EV1), closely resembling cells observed in the proximal and distal abaxial regions of the leaf in *Arabidopsis* (Appendix Fig. S2). In a second scenario, we initially imposed isotropic growth followed by a brief period of vertical anisotropic growth (~5/6 isotropic and 1/6 anisotropic), which led to vertically oriented lobes (Fig. 1E; Movie EV2). Conversely, reversing the growth sequence, starting with anisotropic growth followed by isotropic growth, resulted in horizontally biased lobes (Fig. 1F; Movie EV3). These simulations indicate that differences in the temporal dynamics of growth anisotropy, even under conditions of identical cumulative growth, can lead to significant variations in cell shape. Furthermore, isotropic tissue growth tends

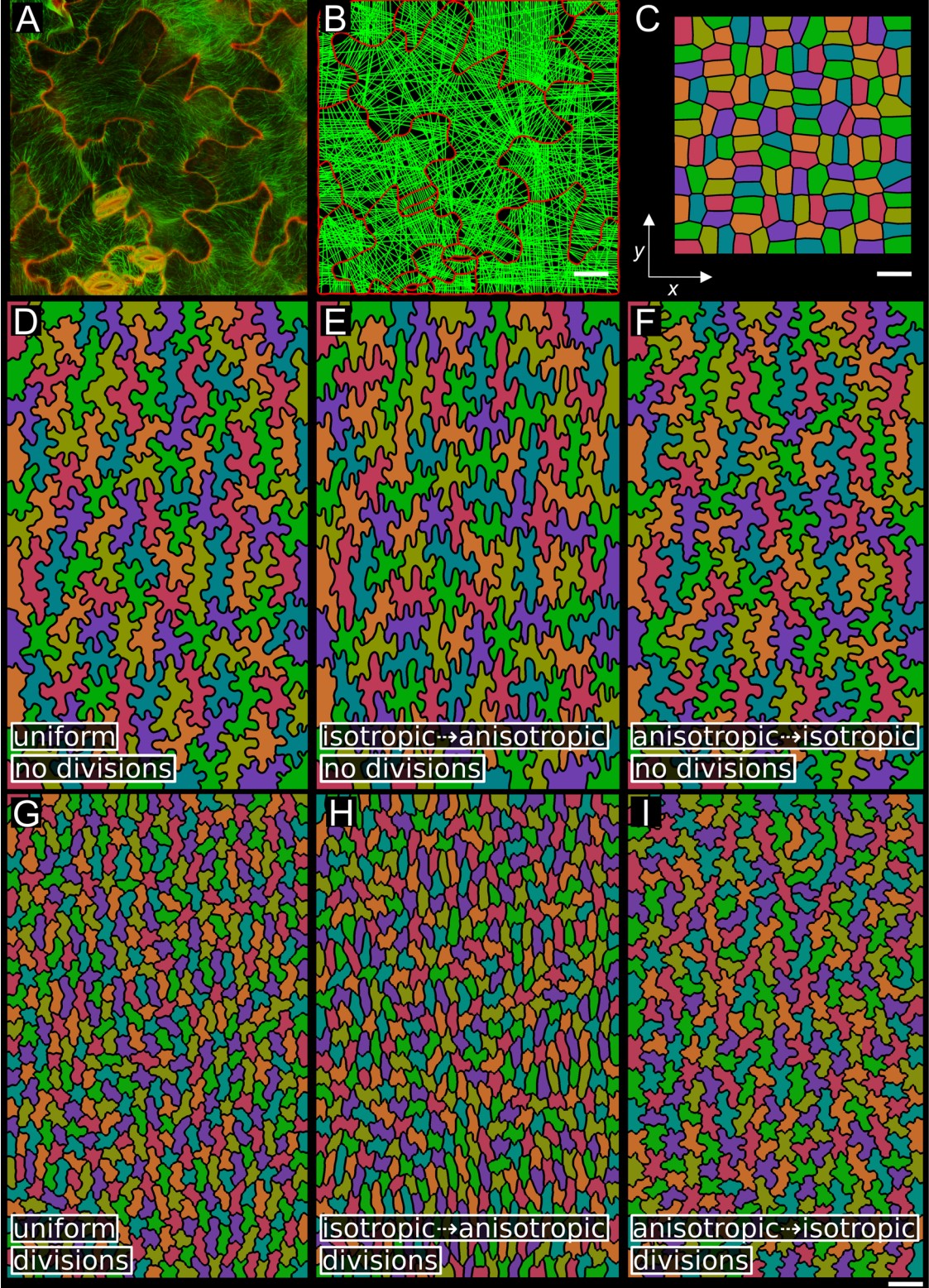

**Figure 1. Growth distribution over time influences the shapes of pavement cells.**

(A) Confocal image representing epidermal pavement cells from a fully developed 3-week-old leaf of *Arabidopsis thaliana*. The cell wall was visualized using propidium iodide staining (red), and the microtubules were visualized using fluorescent GFP-TUA6 (green). (B) A visualization of the placement of connections across the cells for the cell shapes in A in the computational model. (C–F) Model simulation of puzzle cell formation across various growth fields. (C) The initial template. (D–F) Cell shapes evolved over 300 growth iterations to the same final template size, 16 times larger in y and 10 times larger in x than the initial template. (D) Uniform weakly anisotropic growth with puzzle-shaped cells. Growth rates in the x and y directions are different but constant over time. (E) Initially isotropic growth followed by a short period of anisotropic growth in the y direction only (5/6 vs 1/6 of the simulation time), resulting in puzzle-shaped cells whose lobes are largely oriented vertically. (F) The same simulation with the growth reversed, a short burst of anisotropic growth followed by more isotropic growth, resulting in puzzle-shaped cells with a subtle bias in the horizontal orientation. (G–I) Same as (D–F) respectively, with divisions implemented that cease after the first 20 of 100 growth steps. Scale bars, A, B 20 μm. Template scales, C 1x, D–I 16x.

to yield more complex, puzzle-like cell morphologies, while anisotropic growth produces simpler, elongated forms. These observations suggest that final cell shape reflects the cumulative outcome of dynamically evolving growth patterns, with changes in local growth anisotropy influencing morphogenesis over time. This provides insight into how organ and cell shape interact when cells do not have simple shapes. We next turned to the role of proliferation, asking whether continued or halted division alters final cell size and lobing patterns.

## The interaction of cell division and growth

If the purpose of puzzle-shaped cells is to mitigate stress as the cells become large, one strategy would be for the plant cells to keep dividing. Some species appear to do this, making simple polygonal cell shapes (Liu et al, 2021; Remmler and Rolland-Lagan, 2012; Sapala et al, 2018; Vőfély et al, 2019). In plants that do make puzzle cell shapes, different developmental contexts can make different cell shapes, even if the growth dynamics are the same. One example is the adaxial vs abaxial sides of the *Arabidopsis* leaf, which show differences in puzzle cell shape (Appendix Fig. S2) (Liu et al, 2021; Panteris and Galatis, 2005; Vőfély et al, 2019). To explore the impact of the timing of the cessation of cell division, we ran the models with different growth dynamics (Fig. 1D–F) but allowed cell division for the initial 20% of the simulation (Fig. 1G–I; Movies EV4–6). Although not a large proportion of the simulation time, it has a substantial effect on the final cell shape. As expected, more divisions result in smaller and overall, less lobed cells. Less intuitively, it also affects the bias in lobing, which in some cases increases. In the simulation with first isotropic then anisotropic growth, the lobes appear almost exclusively in the vertical direction (compare Fig. 1E,H). In the uniform case, the model approximates the differences in cell shape between the abaxial and adaxial sides of an *Arabidopsis* leaf (compare Fig. 1D,G with Appendix Fig. S2). This suggests that the difference in cell lobing between the abaxial and adaxial sides of the leaf could be explained by the timing of the entry into the endoreduplication cycle.

Experimental manipulation of division timing supports the model's prediction that increased proliferation reduces lobeyness. The LOSS OF GIANT CELLS FROM ORGANS (LGO) regulator of endoreduplication affects this timing (Schwarz and Roeder, 2016). In the *lgo-2* mutant, which undergoes less endoreduplication than Col-0, cells divided more frequently and remained small and less lobed (Appendix Fig. S5A,B). In contrast, LGO overexpression produced larger cells with pronounced lobes (Appendix Fig. S5C). These observations are consistent with the idea that extended proliferation limits the expansion phase, restricting lobe

development, and that changes in lobeyness arise indirectly from altered growth dynamics downstream of endoreduplication (Appendix Fig. S5D). Faster division is also expected to shorten the time available for temporally varying growth anisotropy to shape cell outlines, consistent with the temporal effects in our simulations. While these *Arabidopsis* models reveal how cell division timing influences lobing under different growth patterns, we next asked whether similar principles apply in a species with strong directional growth.

## Experimental validation that cell shape records growth history

We focused on maize (*Zea mays*), whose elongated leaves contain pavement cells with lobes oriented mainly in the transverse direction, an arrangement that is unexpected given the leaf shape, which suggests growth anisotropy should favor the longitudinal axis. To determine the growth rates and directions that produce these cell shapes, we used time-lapse experiments that obtained sequential replicas of cell patches (Fig. 2A1-L2; Appendix Fig. S3) on juvenile maize leaves, covering different distances from the intercalary meristem (Fig. 2M). This approach allowed us to reconstruct the morphogenesis of maize pavement cells using two consecutive time points. Our observations revealed that puzzle-like shapes in maize pavement cells emerge from a two-phase growth pattern: an initial phase of strong anisotropic expansion along the leaf's longitudinal axis, followed by a shift to nearly isotropic growth, during which transverse expansion increases relative to longitudinal expansion. To simulate maize cell development, we initialized the model with a representative geometry of young cells (Fig. 2O). By adjusting growth rates and anisotropy based on time-lapse observations, we reproduced the shapes and timing of maize puzzle cells (Fig. 2N,P; Movie EV7). We perturbed the model by reversing the growth sequence, initiating with transverse growth followed by longitudinal growth, while preserving the total amount of growth. This produced cell shapes that had lobes primarily oriented in the opposite direction, along the longitudinal axis of the leaf, different from those observed in maize (Fig. 2Q; Movies EV8 and 9). These findings highlight the critical role of the dynamics of growth fields in determining cell shapes and explain the formation of cells with lobes that are primarily transversely oriented, even if the overall leaf shape is elongated. By integrating empirical observations with computational modeling, we show that the sequence of anisotropic and isotropic growth phases can explain the puzzle cell shapes observed in maize, demonstrating how temporal growth dynamics shape cell morphology in planta.

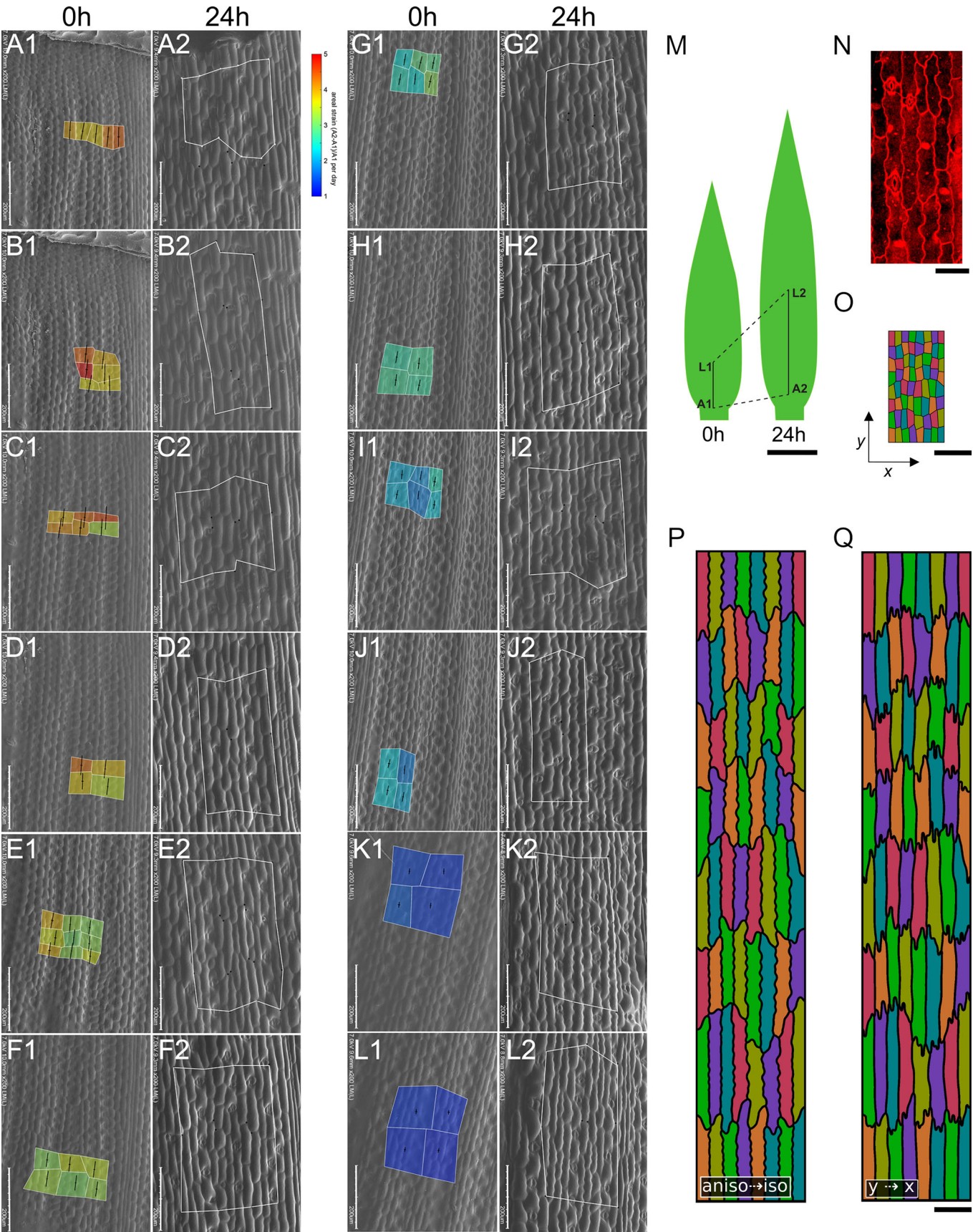

**Figure 2.  Temporal dynamics of maize (*Zea mays*) growth.**

(A1–L2) Electron micrographs show the temporal progression of maize growth and the relative cellular changes after 24 h with images arranged sequentially along the leaf blade from the proximal base (A) to distal regions (L). In each paired set, such as A1 and A2, area strain heatmaps are presented, calculated as $\frac{A2-A1}{A1}$, to visualize the daily growth rate for each region. Distinct boxes in the initial image of every sequence highlight this strain, and the lines within demonstrate the primary growth directions. The distances from the intercalary meristem on the specific blade regions are given: region A is in the meristematic region at 0 h and at circa 200 μm at 24 h; region B at 50 and 430 μm; region C at 185 and 1140 μm; region D at 320 and 1875 μm; region E at 620 and 3655 μm; region F at 760 and 4340 μm; region G at 960 and 5420 μm; region H at 1200 and 6630 μm; region I at 1500 and 7650 μm; region J at 1720 and 8650 μm; region K at 2400 and 10,870 μm; region L at 3100 and 12,730 μm. (M) Schematic drawings of the maize leaf at 0 and 24 h, indicating positions A1 and L1 on the 0 h leaf and A2 and L2 on the 24 h leaf. (N) Cell shape in the maize leaf epidermis under confocal microscopy. (O) Initial starting template of idealized maize cells. (P) Cell shapes that emerge when growth in the model initially follows anisotropic growth rates observed experimentally, then switches to more isotropic growth rates. (Q) The effect on cell shape when the template is first grown in the y direction, then the x direction to reach the same final template size as P. Scale bars, (N) 50 μm, (A1–L2) 200 μm, (M) 10 mm. Template scales, (O) 1x, (P, Q) 16x.

## Effects of disrupted organ growth on puzzle cell formation

Building on the maize results, we tested a specific prediction from the model: bidirectional expansion promotes lobe formation, whereas strong elongation suppresses lobing, even when cumulative growth is comparable. Elongated cells can maintain a small LEC through aspect ratio alone, whereas cells that expand in two directions would otherwise generate larger stress-prone regions unless lobes form. Consistent with this, cells that expand in two directions reduce the LEC by developing lobes, whereas highly elongated, anisotropically growing cells can enlarge while maintaining small LECs and are therefore not expected to develop lobes. To quantify this, we introduced the minimum axis (min-axis), defined as the shortest dimension of the cell, measured by the width of the smallest rectangle that fully encloses it (Fig. 3A). This measure better reflects the aspect of cell size relevant to mechanical stress; long, thin cells have low min-axis values, while cells that are large in both length and width exhibit higher min-axis values, which are expected to be positively correlated with lobeyness. Lobeyness is defined as the ratio of a cell perimeter to the perimeter of its convex hull, the inverse measure of convexity (Sapala et al, 2018) (Fig. 3A). This measure is particularly effective at distinguishing lobed cells from long, thin cells with gentle curves but no pronounced lobes (e.g., boomerang-shaped cells). By integrating the min-axis and lobeyness metrics, we provide a more robust framework for quantifying and distinguishing cell shapes, enabling better insights into the relationship between cell growth dynamics and mechanical stress. Our analysis of *Arabidopsis* revealed a stronger correlation between min-axis and lobeyness than between cell area and lobeyness (Fig. 3B–G), consistent with the idea that combined min-axis and lobeyness better reflect mechanical constraints and stress-mitigation strategies.

To explore how changes in organ shape and growth affect the shape of puzzle cells, we analyzed time-lapse imaging from 6 to 8 DAS in WT Col-0 and in the miR319-overexpressing *jaw-D* mutant (preprint: Harline et al, 2025), which downregulates TCP transcription factors, prolongs cell proliferation, and yields rippled leaves. We extracted multiple complementary metrics (Fig. 4). Heatmaps of lateral (width) expansion ratios between 6–7 and 7–8 DAS (Fig. 4A,B) show that *jaw-D* exhibits enhanced transverse growth compared to Col-0. Cell area heatmaps (Fig. 4C,D) indicate that *jaw-D* cells remain smaller at all stages, and lobeyness heatmaps (Fig. 4E,F) reveal consistently reduced puzzle-shaping in the mutant. Violin plots of $\log_2$ lateral expansion ratios (Fig. 4G) confirm increased width growth in *jaw-D*, while plots of $\log_2$ cell

area (Fig. 4H) and lobeyness (Fig. 4I) confirm that cells are smaller and less lobed. Across three biological replicates, linear mixed-effects analysis of lobeyness detected strong genotype, day, and genotype × day effects (all $P < 0.001$). Lobeyness was similar between genotypes at 6 DAS and 7 DAS, but significantly lower in *jaw-D* by 8 DAS. For cell area, *jaw-D* cells were significantly smaller than Col-0 at all stages (all $p < 10^{-15}$). Proliferation heatmaps between 6–7 and 7–8 DAS (Fig. 4J,K) show more frequent divisions in *jaw-D*, consistent with its higher total cell counts (Fig. 4M). Mixed-effects analysis of cell number revealed a significant genotype × day interaction ($p = 0.047$), with counts diverging significantly only at 8 DAS. Accordingly, *jaw-D* leaves display a higher width-to-length ratio (Fig. 4L), reflecting an altered balance between growth and division at the cellular level.

## Effects of disrupted puzzle cell formation on organ growth

We then asked whether the reverse relationship also holds by analyzing pavement cell morphology in mutants with defective puzzle cell formation and in wild-type plants subjected to pharmacological treatments. In mutants where puzzle shapes were absent, we did not observe cell rupture or death, as reported at the shoot apical meristem (Sapala et al, 2018). One possible explanation is that these cells experience lower mechanical stress, possibly due to changes in the arrangement of neighboring cells and tissue structure. Another hypothesis is that they have compensatory traits such as restricted organ growth, increased cell elongation, or altered cell wall properties. These may reflect developmental adjustments associated with the absence of puzzle-like lobes. Compared to the wild-type *Arabidopsis* (Fig. 5A; Appendix Fig. S4A), the constitutively active Rho-of-Plants 2 (*CA-ROP2*) mutant, which alters actin filament accumulation, displayed non-lobed cells (Qiu et al, 2002). While lobeyness was abolished in *CA-ROP2*, the lack of lobes was partially compensated by elongated shapes (Fig. 5B; Appendix Fig. S4B). Importantly, *CA-ROP2* mutants exhibited relatively mild reductions in overall plant growth, maintaining taller stature due to their more anisotropic growth habit, suggesting that growth effects in this line may be strongest in isotropically growing organs. Similarly, the *anisotropy1* (*any1*) mutant, which carries a mutation in cellulose synthase A1 (CesA1) affecting crystalline cellulose synthesis (Fujita et al, 2013), also lacked lobes but exhibited sinuous cells as a compensatory feature (Fig. 5C; Appendix Fig. S4C). We analyzed the *constitutive triple response* (*ctr1*) mutant, a negative regulator of ethylene signaling, which displayed reduced cell expansion and lobeyness, with smaller cells and smaller leaves

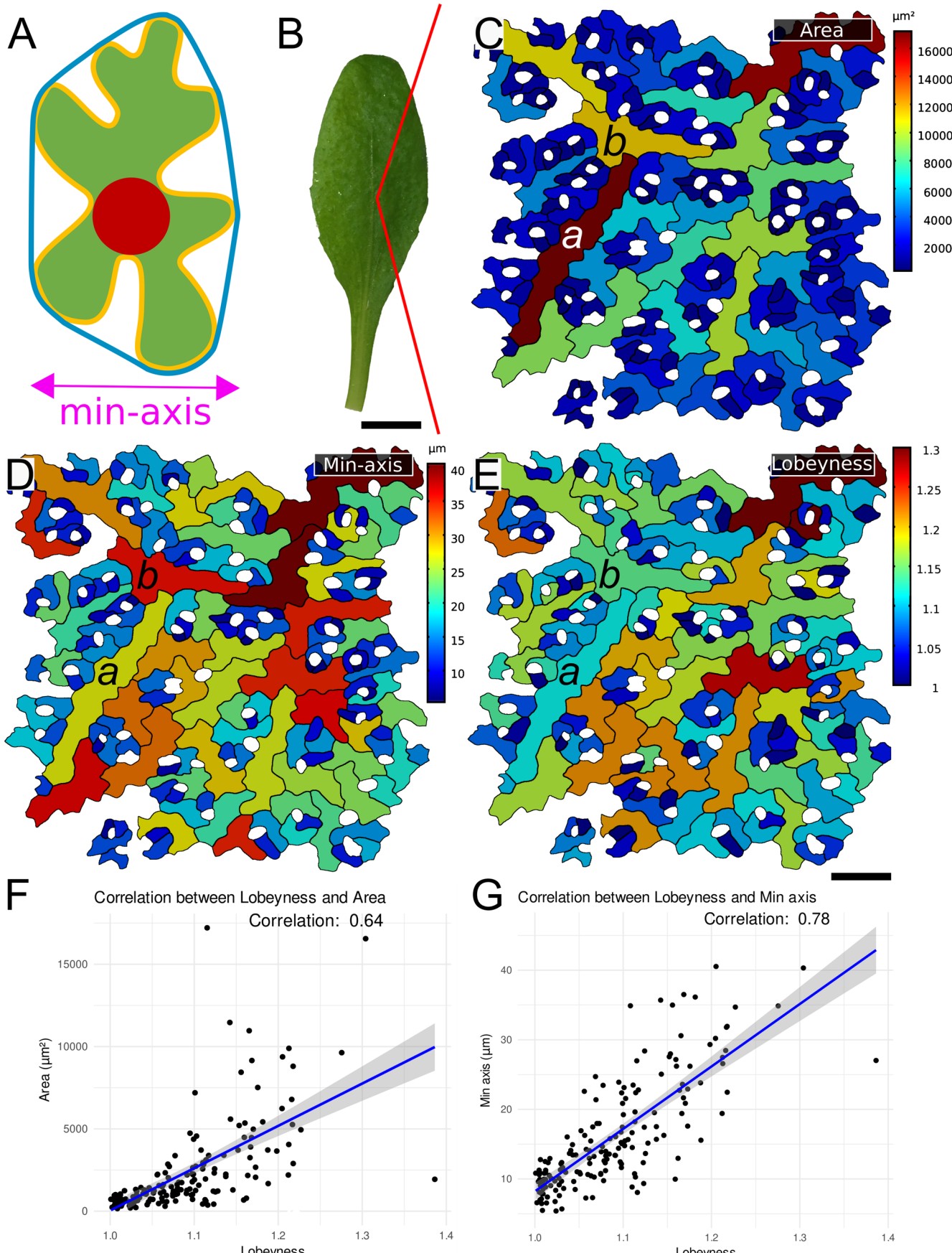

F Correlation between Lobeyness and Area
Correlation: 0.64

G Correlation between Lobeyness and Min axis
Correlation: 0.78

**Figure 3. Lobeyness appears widely as a response to increasing cell size.**

(A) The visual depiction of the pavement cell descriptors: area is shown in green, lobeyness is defined as the cell perimeter (yellow) divided by the perimeter of the convex hull (blue), the min-axis (purple) is defined as the smallest width that would fit the cell, and the largest empty circle (LEC) representing the magnitude of mechanical stress (red). (B) Image of a fully developed 3-week-old leaf of *Arabidopsis thaliana* Col-0. (C–E) Epidermal pavement cells from the leaf in (B), stained with propidium iodide, imaged with confocal microscopy, and segmented with MorphoGraphX. Cell templates are colored by area (C), min-axis (D), and lobeyness (E). (F, G) Graphs presenting the correlation between area and lobeyness (Corr = 0.64) (F) and between min-axis and lobeyness (G) for the same sample as in (C–E). Scale bars, (C–E) 100 μm, (B) 20 mm.

(Fig. 5D; Appendix Fig. S4D). Collectively, these mutants show that reduced lobeyness co-occurs with elongated or zigzag morphologies. These changes could reflect responses to growth constraints but may also result from underlying disruptions in cell wall synthesis, anisotropy, or signaling pathways specific to each mutant background. Without direct functional evidence, the extent to which these altered shapes represent compensatory mechanisms remains unclear. Notably, all mutants with altered lobeyness exhibited weaker correlations between min-axis and lobeyness, along with smaller leaves and overall reduced plant growth compared to wild type (Fig. 5E–H). Together, these mutants show a consistent association between reduced lobeyness, smaller epidermal cells, and reduced plant growth. This trend was also reflected in the 95th percentile of cell lobeyness across genotypes (Fig. 5S). Because these mutations affect diverse processes, these correlations alone do not establish whether reduced lobeyness contributes to the growth defects.

Having shown that reduced lobeyness in *Arabidopsis* mutants correlates with organ growth, we next used pharmacological treatments to acutely disrupt lobe formation in wild-type plants. This allowed us to test whether transient inhibition of lobing, independent of stable genetic mutations, produces similar effects. While maize was used to study how temporal changes in tissue-level anisotropy influence cell shape, *Arabidopsis* cotyledons expand nearly isotropically and are amenable to dose- and time-controlled perturbations of cytoskeletal components. Treatments with oryzalin, a microtubule depolymerizer, and latrunculin B, an actin polymerization inhibitor, were performed on cotyledons to examine the effects of inhibiting lobeyness. By employing dose- and time-dependent treatments, we precisely regulated the inhibitory effects. Compared to control samples (Fig. 5I), low doses of oryzalin led to wider neck regions and larger LECs (Fig. 5J,K). Higher doses further reduced lobeyness, and at very high concentrations, puzzle cell formation was completely abolished, resulting in nearly isodiametric cell shapes (Fig. 5L,M). Similar effects were observed after latrunculin B treatment, which also reduced lobeyness in a dose-dependent manner, although cells retained shallow lobes at intermediate concentrations (Fig. 5N–Q). Quantification of the 95th percentile of lobeyness across treatments confirmed a progressive reduction in lobeyness with increasing drug concentration (Fig. 5R). These treatments produced phenotypic responses similar to those observed in mutants, including reduced cotyledon sizes and overall inhibited plant growth. Although the perturbations used here broadly affect growth and cell wall dynamics in different ways, the consistency of phenotypic outcomes across independent manipulations supports an association between reduced lobing and the observed defects.

## Pavement cell shape reflects the developmental and environmental context of the plant

We have shown that epidermal pavement cell shape depends on the growth dynamics of the organ in which they develop. This raises the question of how often puzzle cells may be absent in one context but present in another with different growth dynamics. To address this question, we examined epidermal pavement cells across various developmental stages, organs, and environmental contexts. Pavement cells exhibit diverse shapes that vary not only across species but also among different organs within the same plant. Their morphology is influenced by developmental stage, environmental conditions, and tissue-specific cues, all of which alter growth dynamics and ultimately determine the extent of lobe formation (Ikematsu et al, 2023; Liu et al, 2021; Majda et al, 2021; Zuch et al, 2022). We first examined how developmental stage affects cell shape by comparing pavement cells in juvenile and mature leaves (or developing vs fully developed leaves, depending on species). In hybrid aspen (*Populus tremula × tremuloides*) clone T89 (Biswal et al, 2025), juvenile leaves were less lobed than mature leaves (Fig. 6A,D), and the min-axis–lobeyness correlation was stronger in mature leaves (Fig. 6G). A similar effect was observed in lilac (*Syringa vulgaris*, Appendix Fig. S6A,B) and camphor tree (*Camphora officinarum*, Appendix Fig. S6C,D). Notably, in lilac, cells previously reported as non-lobed in Vőfély et al (2019) appeared lobed in our lilac samples, likely reflecting differences in developmental stage or environmental conditions. A likely explanation is that the timing of cell division cessation varied across species, with some plants delaying this process until later developmental stages. We frequently observed the coexistence of lobed and non-lobed cells within the same plant, or even within the same organ. For example, in elderberry (*Sambucus nigra*) leaves (Appendix Fig. S7A) and in the bracts of linden (*Tilia cordata*; Appendix Fig. S7B), cells situated above vascular tissues were elongated, while those above the mesophyll were lobed. Similarly, in early developing *Arabidopsis* leaves, the proliferative base contains isodiametric cells, with puzzle-shaped cells emerging distally in the more differentiated regions where division has stopped (Fox et al, 2018).

We found that pavement cell morphology often differs between the abaxial and adaxial surfaces of the same leaf. In species such as love-lies-bleeding (*Amaranthus caudatus*; Appendix Fig. S8A–C), Peruvian lily (*Alstroemeria aurea*; Appendix Fig. S8D–F), fuchsia (*Fuchsia magellanica*; Appendix Fig. S8G–I), peppermint (*Mentha × piperita*; Appendix Fig. S8J–L), and cigar flower (*Cuphea ignea*; Appendix Fig. S8M–O), the abaxial surface shows more pronounced lobing than the adaxial surface, which is directly exposed to sunlight (Watson, 1942). In the cigar flower, this may result from earlier cessation of cell division, as the more lobed cells on the

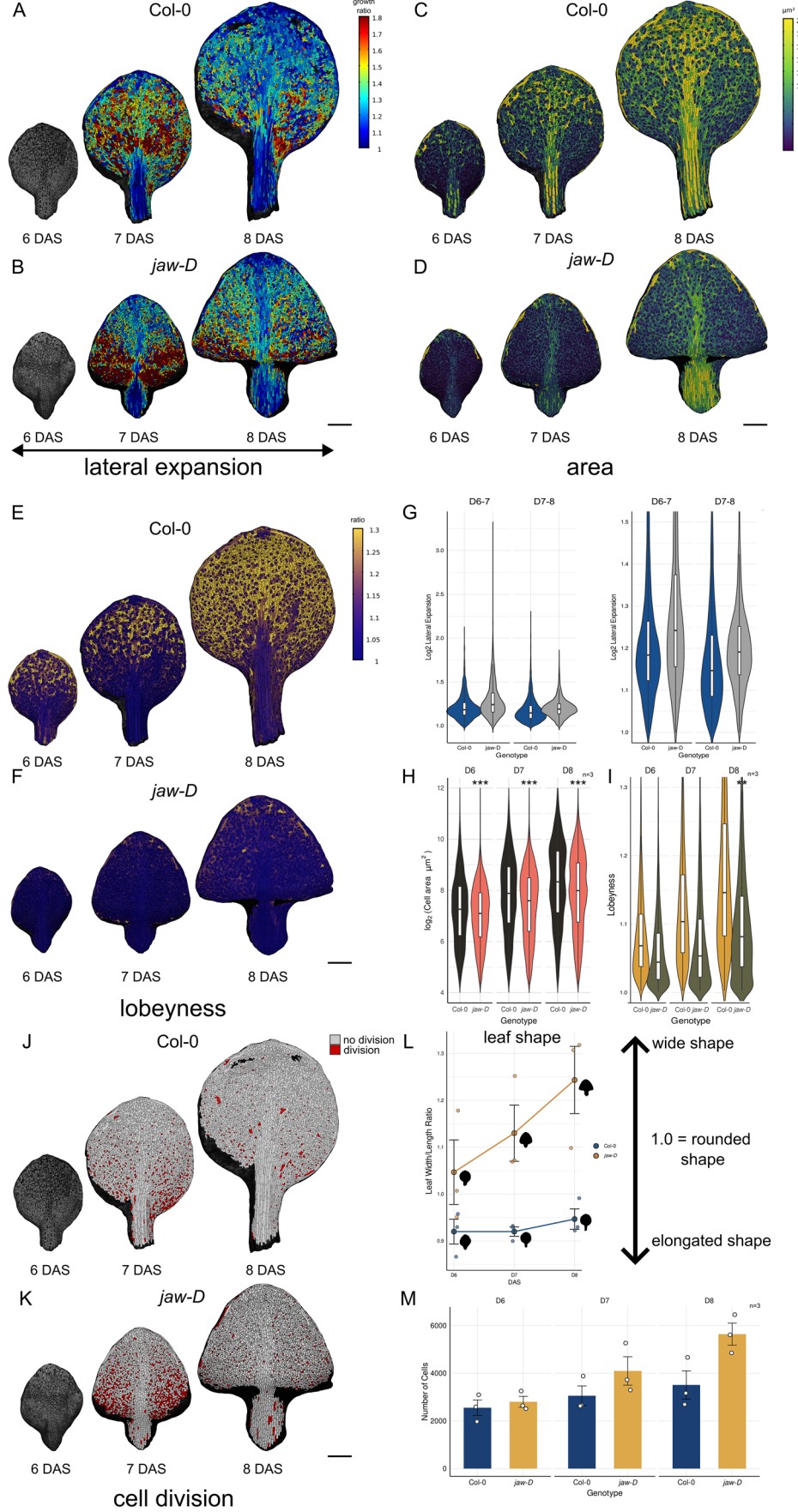

**Figure 4.  Cellular growth, shape and proliferation dynamics in Col-0 and *jaw-D* leaves.**

(A, B) Heatmaps of lateral (width) expansion ratio between 6 and 7 days after sowing (DAS) and 7 and 8 DAS for Col-0 (A) and *jaw-D* (B), with warmer colors indicating greater relative width increase. (C, D) Cell area heatmaps at 6, 7, and 8 DAS for Col-0 (C) and *jaw-D* (D), color-coded by individual cell area. (E,F) Cell lobeyness heatmaps at 6, 7, and 8 DAS for Col-0 (E) and *jaw-D* (F), with warmer hues indicating higher lobeyness. (G) Violin plots with internal boxplots of $\log_2$ lateral expansion ratio from the representative leaves shown in (A, B), grouped by genotype and interval (6–7 and 7–8 DAS). The violin represents the distribution of single-cell values. The center line indicates the median, the box spans the interquartile range (25th–75th percentile), whiskers extend to the most extreme values within 1.5× the interquartile range, and points beyond this range are plotted as outliers. $N = 3404$ (Col-0, 6–7 DAS), 7061 (*jaw-D*, 6–7 DAS), 3714 (Col-0, 7–8 DAS), and 4044 (*jaw-D*, 7–8 DAS). (H) Violin plots with boxplots of $\log_2$ cell area at 6, 7, and 8 DAS ($n = 3$ biological replicates per genotype). The violin represents the distribution of single-cell values pooled across replicates. Boxplots are defined as in (G). Outliers are not shown. Genotype differences are significant at each time point (linear mixed-effects model with Replicate as a random effect; $p < 0.0001$ for 6, 7, and 8 DAS). $N = 7662, 9172, 10527$ (Col-0 at 6, 7, 8 DAS) and 8412, 12,294, 16,926 (*jaw-D* at 6, 7, 8 DAS). (I) Violin plots with boxplots of cell lobeyness at 6, 7, and 8 DAS ($n = 3$ biological replicates per genotype). The violin represents the distribution of single-cell values pooled across replicates. Boxplots are defined as in (G). Outliers are not shown. Genotype differences were not significant at 6 DAS ($p = 0.4803$) or 7 DAS ($p = 0.1054$), but were significant at 8 DAS ($p = 0.0034$) (linear mixed-effects model with replicate as a random effect). $N = 7763, 9285, 10619$ (Col-0 at 6, 7, 8 DAS) and 8573, 12,495, 16,813 (*jaw-D* at 6, 7, 8 DAS). (J, K) Proliferation heatmaps showing the number of cell divisions between 6 and 7 DAS and 7 and 8 DAS in Col-0 (J) and *jaw-D* (K); white indicates no division, red indicates one or more divisions. All heatmaps in panels A–K are derived from the same representative leaf per genotype at each time point, with different cellular parameters visualized from the same sample. (L) Leaf width-to-length ratio over time in Col-0 and *jaw-D* leaves. Individual dots represent biological replicates ($n = 3$ per genotype); circles and lines indicate means ± SEM. A ratio of 1 corresponds to a circular outline, higher values indicate wider leaf shapes, and lower values indicate increased elongation. (M) Bar plot of total cell number per leaf ($n = 3$ biological replicates per genotype), with individual data points shown; error bars represent ±SEM. Asterisks in (M) indicate significance from linear mixed-effects model contrasts: $p < 0.05$ (*), $p < 0.01$ (**), $p < 0.001$ (***). *jaw-D* cells are smaller at all stages (all $p < 10^{-15}$), show significantly reduced lobeyness only at 8 DAS (Col-0: 1.179 ± 0.001; *jaw-D*: 1.099 ± 0.001, $p = 0.003$), and divide more frequently, leading to higher total cell numbers at 8 DAS (Col-0: 1719 ± 40; *jaw-D*: 1912 ± 39, $p = 0.016$). Width-to-length ratios are consistently higher in *jaw-D*, reflecting altered growth balance. Scale bars, (A–F, J, K) 250 μm. Panels (B, D, F, K) are adapted from Harline and Roeder (2023) under a Creative Commons Attribution 4.0 license.

abaxial side are larger. In love-lies-bleeding, fuchsia, and peppermint, this size difference is absent, suggesting variation in the target LEC instead. These patterns highlight the influence of developmental context on cell morphology and indicate that model parameters such as target LEC size can substantially affect cell shape.

Environmental factors also significantly influence pavement cell shape. In camphor tree leaves, typically characterized by non-lobed cells in leaf epidermis (Zhao et al, 2006), cells developed puzzle shapes when grown in greenhouse conditions, with the min-axis-lobeyness correlation of 0.59, whereas this morphology was not observed when plants were cultivated outdoors, with a correlation of 0.3 (Fig. 6B,E,H). Outdoor-grown cells were larger; they remained non-lobed, while smaller indoor-grown cells adopted puzzle-like morphology, again suggesting a difference in target LEC size as a possible cause. The mechanism for how environmental signals could affect the LEC size is not clear, although the observations align with paleoecological studies that use cell sinuosity to distinguish sun and shade leaves and to infer environmental dynamics (Bush et al, 2017; Kürschner, 1997). The data also demonstrate that specific environmental conditions are another case where the appearance of puzzle-shaped cells may be suppressed, even though the species is able to make them.

Next, we wondered if plants that were reported not to make puzzle-shaped cells in leaves were able to make them in other organs. To this end, we examined pavement cell shapes in petals, sepals, and fruits. In certain species, such as linden (*Tilia cordata*; Fig. 6C) and crownvetch (*Securigera varia*; Appendix Fig. S9A), the epidermal pavement cells on the adaxial side of mature leaves were non-lobed. However, despite the absence of lobes in leaves, lobed cells did appear in other organs, for example, in the bracts of linden (Fig. 6F) or the petals of crownvetch (Appendix Fig. S9B). Interestingly, in linden, elongated bracts exhibited lobed cells, whereas rounder leaves did not (Fig. 6I). Species without puzzle-shaped cells in leaves but with them in other organs were not rare (Appendix Fig. S9C–L), with puzzle cell shapes appearing in petals in cemetery iris (*Iris albicans*) and spring crocus (*Crocus vernus*),

and sepals in St John's wort (*Hypericum perforatum*), common gorse (*Ulex europaeus*) and blackthorn (*Prunus spinosa*). These results show that although these plants do not make puzzle-shaped cells in leaves, the mechanism to make puzzle cells is nevertheless present. Across all contexts examined in Fig. 6, the lobeyness distributions varied substantially across developmental stages, environments, and organs (Fig. 6J).

## Prevalence of the mechanism to make puzzle-shaped cells

Because plant organs typically contain a mixture of lobed and non-lobed cells, the average cell shape may not capture the capacity of a species to produce puzzle-shaped cells. To address this, we determined the ability of each species to develop lobed cells by using the 95th percentile of cell lobeyness, which represents the upper range of lobing across different organs and developmental contexts (Fig. 7B,C). We used the 95th percentile rather than the maximum to reduce sensitivity to rare outliers, including occasional tracing or segmentation artifacts. The rationale is that if the plant can make lobed cells, it must possess the mechanism, even if most of its cells are non-lobed. This metric revealed that most species possess the capacity to generate highly lobed cells, even if the average cell shape does not exhibit significant lobeyness as previously reported by Vőfély et al (2019).

To assess the universality of lobe formation in response to increasing cell size, we conducted a large-scale analysis of the correlation between the min-axis and lobeyness across vascular plant species (Fig. 7A). We analyzed cell shapes in 327 species and 663 species-organ combinations, totaling 72,026 cells (Table EV1). In our analysis, we collected samples from 45 species in various plant organs and incorporated data from previously published analysis of leaf epidermal pavement cells in 19 species Sapala et al (2018), 250 species from Vőfély et al (2019), and data from 13 early Miocene species (Reichgelt et al, 2020). In 235 of 327 species (72%), the correlation coefficient between min-axis and lobeyness was $r \geq 0.3$, with particularly high correlations found in tobacco

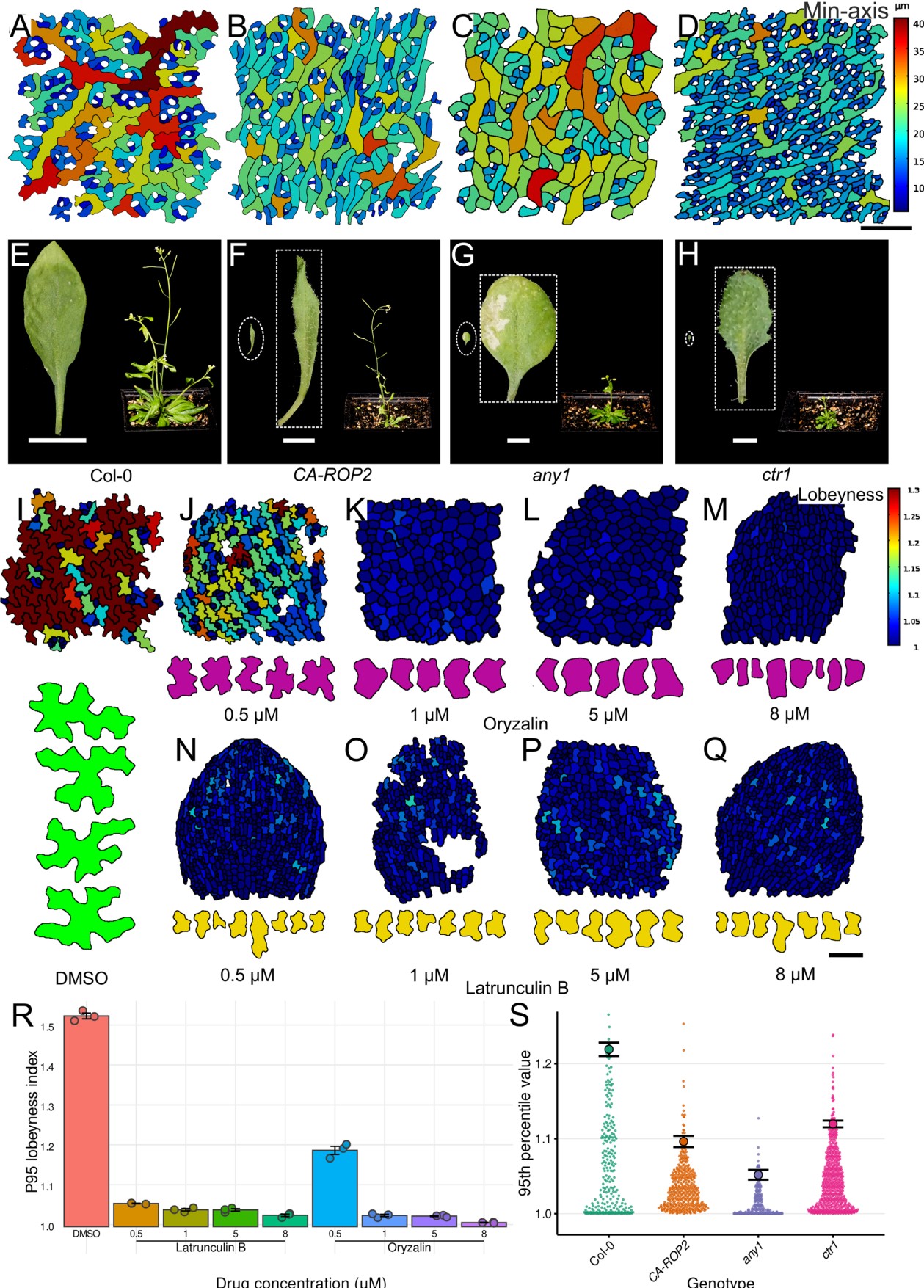

**Figure 5. The lack of lobes is associated with growth defects.**

(A–H) The shape of epidermal pavement cells (A–D) and overall growth phenotypes (E–H) in 3-week-old *Arabidopsis thaliana* Col-0 and mutants. (A, E) Puzzle cells in Col-0 displaying high lobeness and normal growth. (B, F) Thin and elongated epidermal pavement cells in the *CA-ROP2* mutant displaying decreased lobeness (B), with long and thin leaves, and reduced overall plant growth (F). (C, G) Snaky epidermal pavement cells in the *any1* mutant displaying decreased lobeness (C), with smaller and round leaves, and dwarf plant phenotypes (G). (D, H) Small epidermal pavement cells in the *ctr1* mutant displaying decreased lobeness (D), with very small leaves, and dwarf plant phenotypes (H). (E–H) Each panel shows an image of the full leaf on the left and the pot with the full plant on the right; for mutants (F–H), the leaf within the dashed oval is scaled to match (E), while the dashed square provides a zoomed-in view with a separate scale bar. (I–Q) The effect of pharmacological treatments on the shape of epidermal pavement cells. (I) The mock treatment (DMSO) retains the typical morphology of puzzle cells. (J–M) Dose-dependent oryzalin treatments lead to a gradual decrease in lobeness and an increase in LEC (cells labeled in violet). (N–Q) Dose-dependent latrunculin B treatments lead to a decrease in lobeness with shallow lobes (cells labeled in yellow). (R) Treatments were administered at 0.5, 1, 5, and 8 µM. The 95th percentile of lobeness is plotted for each drug treatment. Each dot represents one biological replicate (n = 3 per condition), with values computed as the 95th percentile across multiple pavement cells within each replicate. Bars indicate the mean across replicates and error bars indicate ±SEM. (S) The 95th percentile of lobeness for different mutants. Dots represent individual pavement cell lobeness measurements. Filled circles indicate the observed 95th percentile for each genotype. Bars indicate the 95th percentile for each genotype. Error bars show the bootstrap standard error of the 95th percentile (10,000 bootstrap resamples). Scale bars: (I–Q, contours) 50 µm, (I–Q, epidermis) 100 µm, (A–D) 200 µm, (H, dashed square) 1 mm, (G, dashed square) 2 mm, (F, dashed square) 5 mm, (E, dashed square, F–H, dashed circle) 50 mm.

(*Nicotiana tabacum*, Corr = 0.96), cigar plant (*Cuphea ignea*, Corr = 0.90), morning glory (*Ipomoea tricolor*, Corr = 0.91), black nightshade (*Solanum nigrum*, Corr = 0.90), Java fern (*Microsorum pteropus*, Corr = 0.85), and Peruvian lily (*Alstroemeria aurea*, Corr = 0.89) (Appendix Figs. S10A–F and S11A). Overall, these findings support the hypothesis that the puzzle cell shape is a widespread adaptive strategy for managing mechanical stress during growth.

However, some species displayed weak or even negative correlations between the min-axis and lobeness (Fig. 7A), which seems contradictory to the notion that lobes form in response to a developmental constraint based on stress. Upon further examination of these cases, we identified several reasons for the lack of a strong correlation in certain samples. In cardboard palm (*Zamia furfuracea*), the highly anisotropic growth led to elongated cells without an increase in the min-axis, thus not favoring lobe formation (Appendix Figs. S10H and S11B). Conversely, species like yellow guava (*Psidium guajava*) and pinwheel flower (*Tabernaemontana divaricata*) maintained a small uniform cell size, suggesting frequent divisions to maintain size, even though the leaves can become quite large (Appendix Figs. S10G,J and S11C,D). Weak correlations were also seen in some species with highly lobed cells, such as hare's foot fern (*Davallia solida*; Appendix Figs. S10I and S11E). The min-axis does not capture this case well, as substantial variability comes from branching rather than an increase in lobes, as the cells are all highly lobed. Here, it would be better to sample a patch of the leaf where the cells have a gradient from smaller dividing cells to mature expanded ones. In species without lobes, like agapanthus (*Agapanthus praecox*), cell staggering had a greater impact on thinner, less elongated cells (Appendix Figs. S10L and S11F), whereas in Virginia spiderwort (*Tradescantia virginiana*), irregular shapes in smaller stomatal lineage cells increased the tissue's lobeness scores (Appendix Figs. S10K and S11G). Overall, our analysis shows that most species with lobed pavement cells displayed a significant positive correlation between lobeness and min-axis. These correlations may be underestimated in species for which our samples did not capture the full range from simple to highly lobed cells.

We hypothesize that puzzle cell shape reduces tensile stress in enlarging cells by limiting the largest empty circle (LEC). Consequently, large cells are expected to exhibit a smaller increase in LEC area for the same increase in cell area. To test this prediction, we plotted LEC area versus cell area and fit a quadratic

constrained to pass through the origin (Appendix Fig. S12A). If our hypothesis holds, the relationship should be concave downward (α < 0), indicating that LEC area increases more slowly than cell area. We observed this pattern (Appendix Fig. S12B,C). Moreover, because isotropic enlargement without shape change would tend to increase the available free span, limiting LEC as cells widen requires increased anisotropy through elongation, lobing, or both. Using the min-axis as a proxy for cell width, we found that LEC diameter increases more slowly than the min-axis (Appendix Fig. S12D,E), a relationship that persisted across multiple clades and species (Appendix Fig. S12F–K). Overall, 87% of species (286 of 327) exhibited negative α values (Appendix Fig. S13). An exact binomial test rejects the null expectation that positive and negative α values are equally frequent (two-sided $p = 2.32 \times 10^{-46}$). Negative α values are consistent with LEC increasing more slowly than cell width. Together, these results show a consistent association between increased lobing and reduced LEC expansion relative to growth, supporting the idea that lobing may help limit wall free span and reduce tensile stress.

## Discussion

Since the epidermis plays a pivotal role in shaping plant organ architecture (Beauzamy et al, 2015; Kutschera and Niklas, 2007), it is essential that its cells possess robust mechanisms to mitigate mechanical stress. Here, we show that puzzle cell shapes emerge as a result of the tight interaction between mechanical constraints and growth at the organ scale. Our results are consistent with a stress-buffering function of lobes, but the analyses do not by themselves distinguish an active stress-buffering role from shape emergence as a consequence of mechanical constraints. Our analysis of epidermal pavement cells in maize leaves and computational modeling underscore the critical role of specific growth trajectories in shaping puzzle cells, suggesting that the morphology of these cells may offer insights into the growth history and environmental adaptations of both living and extinct plant species (Couturier et al, 2009). In many cases, the observed variation in the presence or absence of puzzle cells across different contexts likely reflects differences in growth and proliferation dynamics rather than fundamental differences in the mechanism of puzzle cell patterning. In *jaw-D*, time-lapse heatmaps reveal a clear increase in transverse (lateral) expansion, concurrent with smaller, less-lobed cells and

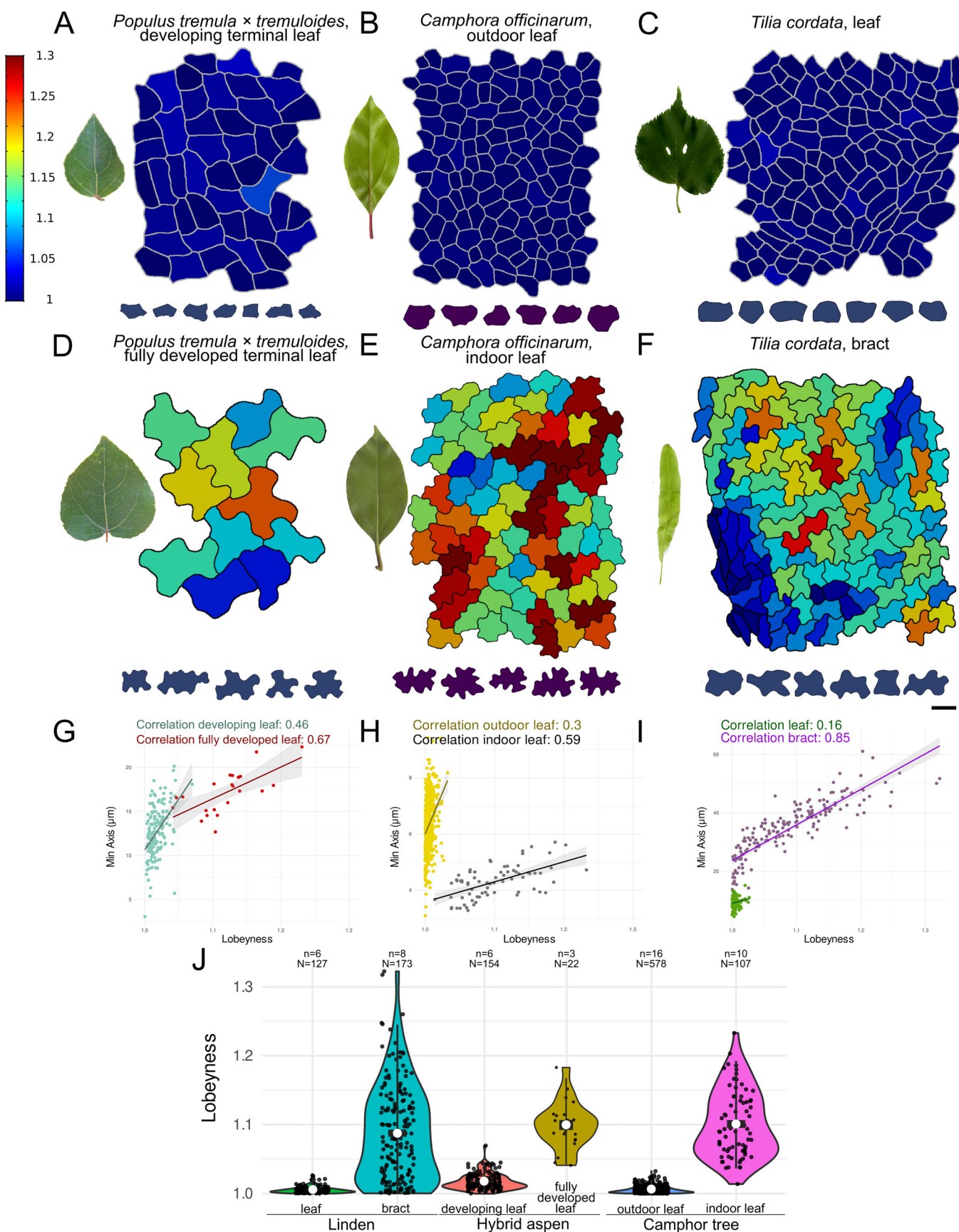

◄

**Figure 6. Pavement cell shapes exhibit developmental and environmental plasticity, varying across organs.**

(A–F) Leaf photographs are paired with lobeyness heatmaps of pavement cell contours and outlines near the 95th percentile of lobeyness. (A, D) Hybrid aspen (*Populus tremula × tremuloides*) clone T89, showing developing (A) and fully developed (D) terminal leaves from juvenile greenhouse-grown trees on the adaxial surface. (B, E) Camphor tree (*Camphora officinarum*) leaves grown outdoors (B) and indoors (E). (C, F) Linden (*Tilia cordata*) leaves (C) and bracts (F). (G–I) Scatter plots of the correlation between lobeyness and min-axis (the shortest cell dimension): (G) hybrid aspen developing (Corr = 0.46) vs fully developed (Corr = 0.67) terminal leaves, (H) camphor tree outdoor (Corr = 0.3) vs indoor (Corr = 0.59) leaves, and (I) linden leaf (Corr = 0.16) vs bract (Corr = 0.85). (J) Violin plots showing lobeyness distributions for each condition or organ. Dots represent individual pavement cells pooled across biological samples. The white circle indicates the median, the black box indicates the interquartile range, and whiskers extend to 1.5× the interquartile range. n indicates the number of biological samples, and N indicates the number of cells for each group. Scale bar: (D heatmap, E heatmap) 10 µm, (A heatmap) 15 µm, (B–D contours) 20 µm, (B heatmap, E contours) 25 µm, (C heatmap) 40 µm, (F heatmap, contours) 50 µm.

elevated division rates; these cellular dynamics map precisely onto the broader, more triangular leaf form. Thus, the puzzle shape phenotype in *jaw-D* is more likely related to its effect on cell proliferation and growth, rather than direct involvement in puzzle cell patterning. This implies that when studying genes that have small to moderate effects on puzzle cell shape, it is necessary to consider that any gene influencing growth rate, timing, direction-ality, or cell division would be expected to influence puzzle cell shape. Differences in puzzle cell shape could reflect growth differences, or differences in cell division timing, and must be excluded before proposing a role in the lobe patterning process itself (Xu et al, 2010).

Genetic modifications in *Arabidopsis* that affect growth anisotropy have puzzle cell phenotypes. Examples are the *ftsh4* mutant that makes sepals nearly isotropic due to altered ROS levels. In this mutant, increased isotropic growth coincides with giant sepal cells adopting puzzle shapes. Conversely, *LONGIFOLIA1* overexpression is associated with long, thin leaves with elongated cells and reduced lobing (Lee et al., 2006; Sapala et al, 2018). Genes that promote cell proliferation lead to the formation of smaller, less lobed cells, as observed in the *lgo-2* mutant (Appendix Fig. S5B,D). On the contrary, genes that inhibit cell division and promote cell expansion, such as those involved in endoreduplication, result in larger, more lobed cells, as in the *LGO OX* line (Appendix Fig. S5C,D). This intertwining of growth dynamics and cell prolifera-tion with the development of puzzle cell shapes highlights the importance of distinguishing between genes specifically affecting the mechanisms of puzzle cell formation from those controlling growth amount, growth anisotropy, and cell division.

Our findings are consistent with the idea that the underlying mechanism involves microtubule-guided cellulose deposition that patterns local wall anisotropy and biases growth during lobe initiation and outgrowth. Pharmacological treatments using oryzalin and latrunculin B are consistent with a requirement for intact microtubules and actin-dependent trafficking during lobe formation. Oryzalin depolymerizes microtubules and reduces lobe initiation during the treatment window, whereas latrunculin B broadly perturbs actin-dependent trafficking and secretion that support cell wall remodeling during polarized expansion (Bidhendi et al, 2019; Panteris and Galatis, 2005; Qiu et al, 2002). Pre-existing lobes can persist after acute cytoskeletal disruption, so these treatments mainly constrain new lobe formation or further amplification rather than erasing established geometry. Together, these results are consistent with models in which microtubule-guided wall patterning and mechanical coupling between neighbor-ing cells drive the interdigitated puzzle shape, as opposed to models that attribute lobe formation primarily to purely cell-autonomous protrusive outgrowth (Belteton et al, 2021; Sapala et al, 2018).

*Arabidopsis* mutants unable to form puzzle cells exhibit stunted growth, consistent with a role for puzzle cell shape in facilitating cell expansion during normal development. While some plants developed snaky cells as a compensatory response to the absence of puzzle cells, this alternative morphology does not fully restore normal organ growth and is rarely observed in both extant and fossil records, suggesting that it may be evolutionarily disfavored compared to the more common and adaptable puzzle cell morphology. Collectively, these observations support the view that puzzle cell formation emerges as a developmental response to mitigate the high mechanical stresses that would be created in large isodiametric cells, and that it is mediated by growth restriction, a process that is likely broadly conserved in vascular plants.

Despite the apparent complexity of factors influencing puzzle cell formation, their widespread emergence reflects a response to a physical constraint shared by all plants. Puzzle cell formation is a common strategy in organs with large, isotropically growing cells, where it helps prevent the formation of extensive, unreinforced areas that would otherwise be exposed to high mechanical stress. However, puzzle cell shape is only required when cells become large in more than one direction, and multiple strategies may mitigate this need. For example, some species, such as yellow guava (*Psidium guajava*), appear to maintain smaller, more isodiametric epidermal cells, potentially through higher division rates, which would limit cell enlargement and thus lessen the mechanical demand for lobing. Alternatively, many plants transition from actively dividing, meristematic cells to larger, endoreduplicated cells (Appendix Fig. S5), a process common in the epidermis of leaves, roots, and sepals of *Arabidopsis*, that often promotes puzzle cell formation as cells expand. In roots and hypocotyls, growth is highly anisotropic, resulting in large but elongated cells; in these cases, the elongated shape minimizes open areas and makes puzzle cell formation unnecessary. However, in organs where growth is less anisotropic, such as leaves and cotyledons, larger cells adopt the puzzle shape to mitigate bulging and mechanical stress. Under isotropic growth conditions, the absence of puzzle cells would lead to the development of large, unreinforced regions in the periclinal cell wall, exposing cells to considerable turgor-induced stress. Another strategy to reduce stress in large, nearly isodiametric epidermal cells is increased investment in the outer epidermal wall, sometimes accompanied by a multilayered epidermis or hypoder-mis with thickened outer walls (Galletti et al, 2016). Such xeromorphic architectures have been described in leaves of oleander (*Nerium oleander*) and common fig (*Ficus carica*), which show a thick cuticle and additional epidermal or hypodermal layers that reinforce the leaf surface (Abdalla et al, 2016; Bercu and Popoviciu, 2014). Leaves of *Peperomia obtusifolia* likewise contain a multilayered hypodermis with localized wall thickening at the

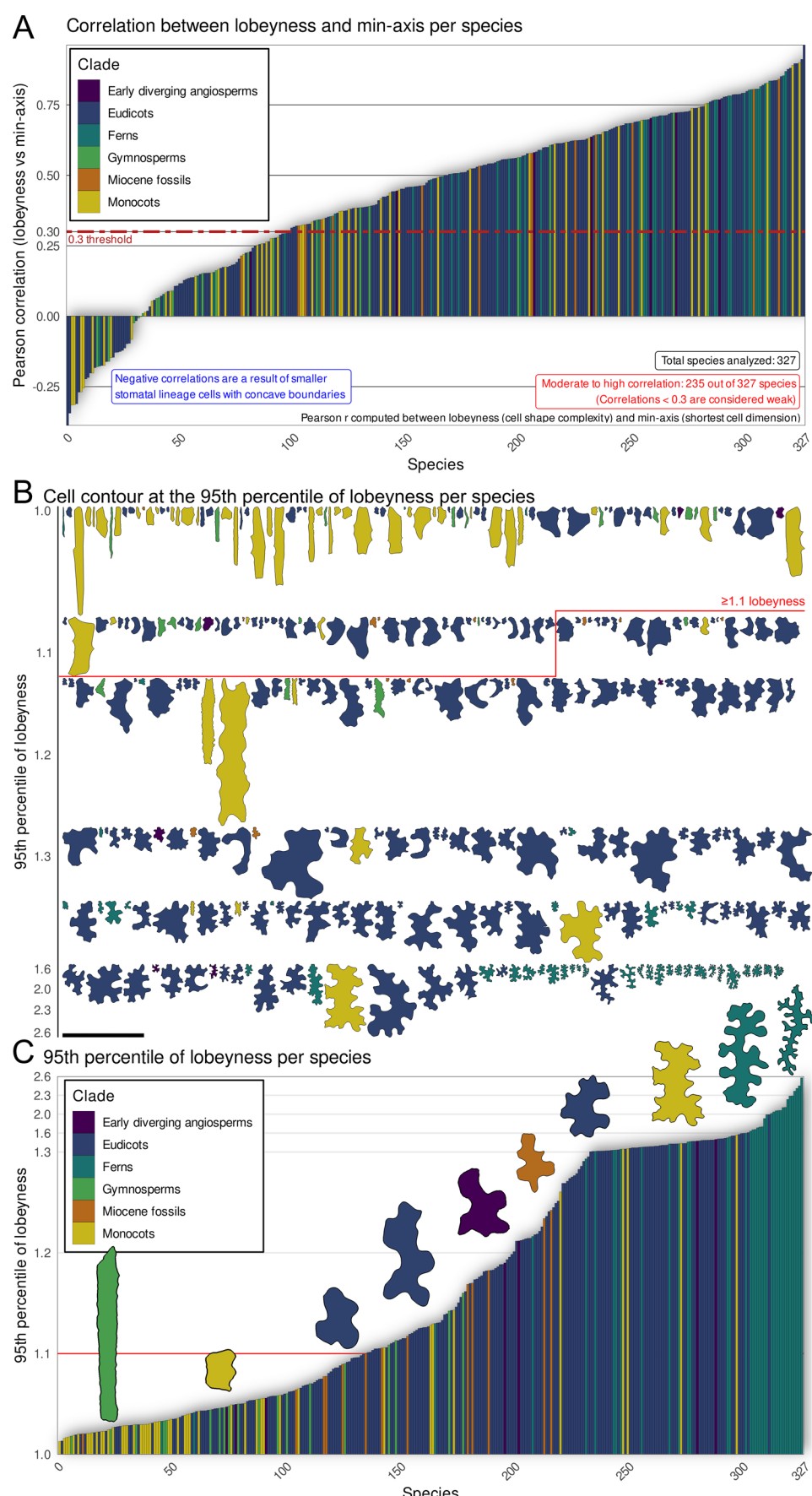

**A** Correlation between lobeyness and min-axis per species

**B** Cell contour at the 95th percentile of lobeyness per species

**C** 95th percentile of lobeyness per species

◀ **Figure 7. Lobeyness is a common feature observed across different species.**

(A) Bar plot displaying the Pearson correlation (r) computed between lobeyness (cell shape complexity) and min-axis (the shortest cell dimension) across 327 species. Correlations below 0.3 are deemed weak, while those at or above 0.3 are classified as moderate to high. In our analysis, 235 out of 327 species exhibited moderate to high correlations. The 0.3 threshold was selected as a heuristic cutoff to differentiate species with pronounced positive relationships from those with less pronounced associations. Bars are color-coded by clade. (B) Visual representation of a single pavement cell contour per species, selected at the 95th percentile of lobeyness. The contours are arranged in order of increasing cell shape complexity, color-coded by clade. The 1.1 threshold was selected as a heuristic cutoff to differentiate species with significant lobeyness from those with less pronounced lobes. (C) Bar plot of the 95th percentile of lobeyness values with selected cell contours superimposed. The contours are color-coded by clade. The contours at approximate lobeyness values 1 (mountain cypress, *Widdringtonia nodiflora*), 1.05 (Panama hat plant, *Carludovica palmata*), 1.1 (yellow trumpetbush, *Tecoma stans*), 1.15 (garlic mustard, *Alliaria petiolata*), 1.2 (Japanese star anise, *Illicium anisatum*), 1.25 (colicwood, *Myrsine*), 1.3 (yarrow, *Achillea millefolium*), 1.5 (Peruvian lily, *Alstroemeria aurea*), 2 (*Phanerophlebia*), and 2.5 (rough maidenhair fern, *Adiantum hispidulum*) were selected to represent all clades at increasing positions along the lobeyness gradient (minimum 1.0 to maximum 2.6). Emphasis is given to the range from 1.0 to 1.3, as significant lobeyness begins to appear around a lobeyness value of 1.1. Scale bars: (C contours lobeyness values 1, 1.05, 1.15, 1.25, 1.3, 2, 2.5) 100 μm, (C contours lobeyness values 1.1, 1.2) 200 μm, (C contours lobeyness value 1.5) 400 μm, (B) 500 μm.

hypodermis to mesophyll interface (Horner, 2012). Compared with shaping the periclinal surface through lobing, these strategies likely require greater cell wall deposition and therefore higher material investment (Majda et al, 2017; Majda et al, 2019). Interactions with subepidermal tissues may also contribute to epidermal lobing. The mesophyll contains air spaces that alter local mechanical support, and differences in tissue curvature between adaxial and abaxial surfaces could influence the distribution of wall stress. Such factors may help explain why lobing differs between the two surfaces in *Arabidopsis* and other species. Although direct experimental evidence remains limited, these interactions represent an important frontier for understanding how epidermal and subepidermal layers coordinate during organ growth. Our large-scale analysis of pavement cells across vascular plants, including plant organs beyond leaves such as sepals, bracts, and even fossil species (Reichgelt et al, 2020), showed that most species contain cells that exhibit significant puzzle cell shapes (Fig. 7), whereas species lacking puzzle cells tend to have elongated cells or cells with small sizes. Interestingly, we found that while some species may not exhibit puzzle cells in their leaves, they can still form them in other organs, such as sepals or bracts, and that the appearance of puzzle cells is influenced by developmental stage and environmental conditions. Overall, our integrated analysis reveals that puzzle cell formation is a dynamic, evolutionarily conserved strategy that enables plants to optimize cell and tissue integrity in response to mechanical constraints, thereby ensuring robust organ development across diverse environmental contexts.

## Methods

### Reagents and tools table

| Reagent/resource | Reference or source | Identifier or catalog number |
|---|---|---|
| **Experimental models** | | |
| *Arabidopsis thaliana* Col-0 ecotype | Nottingham Arabidopsis Stock Center (NASC) | N1092 |
| *Arabidopsis thaliana* anisotropy1 (any1) | Fujita et al, (2013) | N/A |
| *Arabidopsis thaliana* constitutively active GTP-bound ROP2 (CA-ROP2) | Li et al, (2001) | N/A |
| *Arabidopsis thaliana* constitutive triple response1 (ctr1) | Kieber et al, (1993) | N/A |
| *Arabidopsis thaliana* GFP–α-TUBULIN6 (GFP-TUA6) | Ueda et al, (2003) | N/A |
| *Zea mays* cv. 'Polonez' | Commercial cultivar | N/A |
| *Zea mays* cv. 'Cosmo' | Commercial cultivar | N/A |
| Miocene fossil plant specimens (Foulden Maar diatomite core) | Reichgelt et al, (2020), reused in this study | N/A |
| **Antibodies** | | |
| N/A | | |
| **Oligonucleotides and other sequence-based reagents** | | |
| N/A | | |
| **Chemicals, enzymes and other reagents** | | |
| Ethanol (EtOH), 70% | Sigma-Aldrich | Cat: 64-17-5 |
| TWEEN 20 | Sigma-Aldrich | Cat: P9416 |
| Square Petri plates | Sigma-Aldrich | Cat: Z692344 |
| Murashige and Skoog Basal Salt Mixture (1/2 MS) | Duchefa Biochemie | Cat: M5524 |
| Sucrose | Fisher | Cat: 57-50-1 |
| Agar | Duchefa Biochemie | Cat: P1001 |
| Propidium iodide (PI) | Sigma | Cat: 81845 |
| Low melting point agarose (LMP agarose), 3% | ROTH (Germany) | Cat: 39346-81-1 |
| Oryzalin | Sigma-Aldrich | Cat: 36182 |
| Latrunculin B | Sigma-Aldrich | Cat: L5288 |
| Dimethyl sulfoxide (DMSO) | Sigma | Cat: D8418 |
| Take 1 Advanced Impression Material (light body wash) | Kerr Corp., Romulus, USA | N/A |
| Devcon 2 Ton Clear epoxy | Devcon | N/A |
| Hydrogen peroxide | N/A | N/A |

| Reagent/resource | Reference or source | Identifier or catalog number |
|---|---|---|
| Tetrasodium pyrophosphate | N/A | N/A |
| Crystal Violet | N/A | N/A |
| Glycerin jelly | N/A | N/A |
| **Software** | | |
| MorphoGraphX 9 | MorphoGraphX | N/A |
| MorphoDynamX | MorphoDynamX (www.MorphoDynamX.org) | N/A |
| MDXtoR plugin | N/A | N/A |
| Fiji | Fiji | N/A |
| GIMP | GNU Image Manipulation Program (GIMP) | N/A |
| TSView | TSView | v7.1.1.2 |
| ZEN | Zeiss | v2.3 |
| LAS software | Leica | N/A |
| Vlab modeling environment | Algorithmic Botany Virtual Laboratory | N/A |
| funcedit | Algorithmic Botany Virtual Laboratory | N/A |
| **Other** | | |
| Leica STELLARIS upright laser confocal microscope | Leica | N/A |
| Water immersion objective | Leica | 25x/0.95 |
| Zeiss Axio Imager M2 microscope with Axiocam 512 | Zeiss | N/A |
| Nikon Microphot-SA microscope with Leica camera | Nikon/Leica | N/A |
| Nikon Optiphot microscope | Nikon | N/A |
| Hitachi UHR FE-SEM SU8010 | Hitachi | SU8010 |
| Zenodo repository deposit (data, models, MorphoDynamX processes/software) | Zenodo | https://doi.org/10.5281/zenodo.15184042 |
| Plant organ scans (modified from herbarium specimens) | Neuchâtel Herbarium; Meise Botanic Garden Herbarium; Field Museum; Oxford University Herbarium | NEU000004163; BR0000025021905; 304262; 00058444 P; 289029 |

Reagents, experimental models, software, and equipment used in this study are listed in the Reagents and Tools Table.

## Plant material and growth conditions

The seeds were sterilized by immersing them in a solution of 70% EtOH (Sigma-Aldrich, Cat: 64-17-5) and TWEEN® 20 (Sigma-Aldrich, Cat: P9416) for 2 min. Subsequently, they were rinsed twice with 95% EtOH and left to air dry. The sterilized seeds were

then placed in square Petri plates (Sigma-Aldrich, Cat: Z692344) filled halfway with a growth medium composed of ½ MS (Murashige and Skoog Basal Salt Mixture, Duchefa Biochemie, Cat: M5524), 1% sucrose (Fisher, Cat: 57-50-1), at pH 5.6, and 0.7% agar (Duchefa Biochemie, Cat: P1001). After a period of 72 h of stratification at 4 °C, the seeds were grown in a vertical position for 5 days at 22 °C, with a 16-h light cycle per day, in a controlled environment room. Subsequently, the 5-day-old seedlings were transferred to soil and cultivated for 4 weeks at 22 °C, with a 16-h light cycle per day. The screen of epidermal pavement cell outlines was conducted on *Arabidopsis thaliana* Col-0 ecotype and the following lines: *anisotropy1* (*any1*) (Fujita et al, 2013), *constitutively active GTP-bound ROP2* (*CA-ROP2*) (Li et al, 2001), and *constitutive triple response1* (*ctr1*) (Kieber et al, 1993). For the microtubule imaging, seeds of green fluorescent protein α-TUBULIN6 (GFP-TUA6) line (Ueda et al, 2003) were grown on a Petri plate for 12 days and stained with 0.1% PI prior to imaging.

## Confocal imaging of the cell outlines and image processing

The epidermal pavement cell outlines were imaged from the adaxial surfaces of 3-week-old *Arabidopsis thaliana* leaves. Small sections (5 mm$^2$) were sampled from the central part of the leaves and submerged in a staining solution of 0.1% Propidium Iodide (PI) (Sigma, Cat: 81845) dissolved in water for 20 min. After a brief rinse in water, the samples were mounted on a slide with a cover slip and imaged using a Leica STELLARIS upright laser confocal microscope equipped with a water immersion objective (25x/0.95). The excitation and emission wavelength windows used were 488 nm and 600–650 nm for PI and 488 nm and 495–545 nm for GFP, respectively. The images were acquired at $1024 \times 1024$ resolution and z-stack slices of 0.5–1 μm in the z-direction. At least ten different plants and leaves from the adaxial sides were collected at the same developmental stage for each genotype. The confocal images were processed in MorphoGraphX using *Gaussian Blur Stack* and *Brighten Darken* filters to enhance the clarity of cell outlines. The confocal signal was then projected onto a mesh, reconstructing the three-dimensional structure of the sample (2.5D). Cell segmentation was performed using seeding and *Watershed Segmentation* processes. To better capture the cell size aspect relevant to stress, we introduced the min-axis parameter. Unlike cell area, the min-axis accounts for the smallest dimension of the cell by measuring the width of the minimum bounding rectangle (Fig. 3A). This approach allows for a more accurate representation of cell size in the context of lobing, as elongated cells with large areas but small LECs are not expected to form lobes. The min-axis parameter distinguishes between long, thin cells (low min-axis values) and cells that are large in both dimensions (high min-axis values). Lobeyness, calculated as the ratio of a cell's perimeter to its convex hull perimeter (the reciprocal of convexity), was employed to assess lobing. Heatmaps for cell parameters, including lobeyness, min-axis, area, and LEC, were generated using MorphoDynamX. Correlation analysis between contour parameters was conducted using the MDXtoR plugin.

## Sampling of different plant species

The analysis of epidermal cell outlines from different species (see Table EV1) was conducted using specimens collected from commonly grown species in Cologne (Germany) and Norwich (United Kingdom). Plant

material was sampled from non-protected sites where collection was permitted, and no collection permits were required. To obtain the cell outlines, we employed the imprinting method with 3% low melting point (LMP) agarose (ROTH, Germany, Cat: 39346-81-1) dissolved in dH$_2$O. The agarose solution was briefly microwaved until boiling and then allowed to cool to room temperature without solidifying. On a microscope slide, a droplet of the lukewarm agarose was carefully dispensed, and small fragments (1 cm²) were excised from the central region of the plant organs using a razor blade and then gently transferred onto the lukewarm agarose droplets and allowed to solidify. The plant sections were carefully removed from the slide while keeping the agarose intact, thus preserving the shape of the epidermal cells. The slides with the agarose imprints were subsequently stored at 4 °C until they were imaged. Imprints were obtained from both the adaxial and abaxial surfaces where applicable. The identification of plant species was conducted using references such as Simpson (2006) and Harris and Harris (2001). Refer to Sapala et al (2018) and Vőfély et al (2019) for the methods used for pre-existing contours.

## Phase-contrast microscopy of epidermal imprints

The transparent epidermal imprints, with a thickness of 5–10 µm, were visualized using either a Zeiss Axio Imager M2 microscope equipped with an Axiocam 512 camera, or a Nikon Microphot-SA microscope connected to a Leica camera. Air Plan-NEOFLUAR objectives of either x20 or x40 were utilized, depending on the cell size of the sample being imaged. To ensure representative coverage, each image typically contained ~50 cells, adjusted based on the specific sample. The phase-contrast mode was employed for imaging on both microscopes. The images were acquired using ZEN 2.3 software for the Zeiss Axio Imager M2 microscope, utilizing the tiling mode to stitch multiple images (5 × 5), or LAS software for the Nikon Microphot-SA microscope.

## Image acquisition for Miocene fossil plant species

The Miocene plant fossils analyzed in this study were obtained from a dataset published by Reichgelt et al (2020). These fossils were preserved in turbidite deposits within the Foulden Maar diatomite core. To prepare the fossilized leaves for analysis, they were treated with hydrogen peroxide and tetrasodium pyrophosphate salt crystals. This treatment effectively removed mesophyll cell debris, ensuring clearer observation of the leaf structures. The cleaned fossil leaves were then stained with Crystal Violet and mounted on glass slides using glycerin jelly. High-resolution images of the leaves were captured at 100× magnification using TSView 7.1.1.2 microscope imaging software on a Nikon Optiphot microscope. Species identification of the fossil specimens was based on paleobotanical studies and comparison with known species from the Foulden Maar surface exposures.

## Image processing for multispecies pavement cell outline analysis

To improve the visualization of cell outlines, the GNU Image Manipulation Program (GIMP) was used to enhance saturation, contrast, and sharpness. Each image contained a maximum of 50 cells, and only cells with clear outlines were manually traced. Fiji software was used to calculate the distance in pixels along the scale length, which was used to calibrate the scale bars in MorphographX. To aid in cell segmentation, various image filters, including *Invert*, *Gaussian Blur Stack*, and *Brighten Darken*, were applied. The surface mesh was extracted using the marching cubes surface algorithm with a cube size of 5 µm. The cell outlines from the phase-contrast microscopy images were projected onto the mesh, and cell segmentation was achieved by applying the watershed segmentation process. To further refine the cell contours, the smooth mesh process was applied, resulting in a smoother and more accurate representation of the cells. The cell contours obtained from the segmented mesh were then analyzed and quantified using MorphoDynamX. A subset of the most representative cell contours was selected for further analysis, ensuring a focus on the cells that were most relevant to the research objectives. Heatmaps for lobeyness, min-axis, area, and LEC were generated using the contour process. Correlation analysis between two contour parameters was performed using the MDXtoR plugin.

## Dose-dependent drug treatment

To investigate the effects of oryzalin or latrunculin B on *Arabidopsis thaliana* cotyledon pavement cells, plates containing 50 mL of MS medium (0.7% agar, without sugar and vitamins) were prepared. The drugs, oryzalin (Sigma-Aldrich, Cat: 36182) or latrunculin B (Sigma-Aldrich, Cat: L5288), were added to the plates at final concentrations of 0.5, 1, 5, and 8 µM. As a control, DMSO (Dimethyl sulfoxide, Sigma, Cat: D8418) was added to the growth medium at a volume equivalent to that used for the drug treatments. After 6 days, the cotyledons were collected and mounted on microscope slides with cover slips. Images of the cotyledon pavement cells were captured using a Leica STELLARIS upright laser confocal microscope equipped with a water immersion objective (25x/0.95). The captured images were then processed to segment the cells and quantify their contours.

## Growth quantification for the epidermis of juvenile maize leaves

Maize (*Zea mays* cv. 'Polonez' and 'Cosmo') caryopses were kept submerged in water for 6 h, germinated on blotting paper for 18 h in light at room temperature, transferred to plastic containers filled with moist soil, and grown in a glasshouse (temperature 19–21 °C, additional illumination to obtain 16 h day). After 4–7 days, a longitudinal strip of coleoptile or coleoptile and the first leaf sheath was gently removed to expose the base of the first or second juvenile leaf, respectively. The sequential replica method (Kwiatkowska and Burian, 2014; Williams and Green, 1988) was used to obtain silicon molds, made of Take 1 Advanced Impression Material (Light Body Wash, Kerr Corp., Romulus, USA) from the abaxial epidermis of the exposed leaf surface. Two replicas were taken from each leaf at 24 h interval. After the replica taking, the exposed leaf portion was protected from drying by food wrap. Epoxy resin replicas (casts made from Devcon 2 Ton Clear epoxy) were obtained from the silicon molds, sputter-coated and observed using scanning electron microscopy (Hitachi UHR FE-SEM SU8010). Pairs of stereo images were taken from each region of interest and used for the stereoscopic reconstruction of the leaf surface (Routier-Kierzkowska and Kwiatkowska, 2008). Growth was analyzed for two juvenile leaves of cv. 'Polonez' and three juvenile leaves of cv. 'Cosmo'. For each leaf, 3–12 patches of epidermis were identified in the two consecutive replicas, located at

an increasing distance from the intercalary meristem. Because maize pavement cells become strongly elongated, in each patch, 3–9 groups (nearly square-shaped in the first replica), which comprised 6–12 adjacent cells, were used to compute areal growth rates and Principal Growth Directions (Dumais and Kwiatkowska, 2002).

## Contour loading and preprocessing

Contours from Vőfély et al (2019) (https://doi.org/10.5061/dryad.g4q6pv3) were loaded into MorphoDynamX (www.MorphoDynamX.org) using a custom process written for this purpose (Mesh/Contours/Vofely Load Contours). Each cell was given a unique label, and then labelings (one-to-many groupings) were assigned based on clade, species, and species-side. Contours were scaled so that units are in microns and smoothed with a single pass position average of their 1-neighborhoods to reduce noise. A small number of contours were encountered with topological errors, such as cells in multiple pieces or with vertices out of order, causing self-intersection. The very few cells in multiple pieces were removed, and a process was written to select vertices out of order. These specific vertices were smoothed again, and the process was repeated, which removed any remaining problems.

## Contour processes

Several new processes were added to MorphoDynamX to aid in contour processing. These processes are as follows:

*Arrange contours* - Arrange contours in a grid for visualization. This process also sorts them by a specified heatmap, or the current if nothing specified, for example, lobeyness.

*Create contours* - Create contours from a 2.5D surface mesh. This simplifies cells made of multiple faces into a single planar face and rotates them into the XY plane.

*Create LEC* - Create a cell complex with the LEC visualized. The cell label and heat are preserved.

*Load contours* - Loads contours from a directory as text files. Contour files are specified as a list of 2D positions, one per line, with the units assumed to be microns.

*Load contours labeled* - Loads contours from a directory structure root/clade/species/organ/sample.

*Rotate min-axis* - Rotates contours so that the min-axis is in the X direction.

*Save contours labeled* - Saves contours to a directory structure root/clade/species/organ/sample.

*Select bad contours* - Selects points on contours where there are topological problems.

*Select by percentile* - Selects the specified percentile cell for the specified labeling.

*Trim contours* - Remove selected points from contours.

## Min-axis calculation

For a 2D contour, the min-axis is the axis along which the contour has minimum width. To accelerate computations, we use the convex hull of the contour, which substantially reduces the complexity of the contour without affecting its width with respect to a given axis. To find the min-axis, we densely sample axis orientations between 0 and $2\pi$, and project the convex hull onto each axis to determine the contour's width. Once the min-axis has been identified, we set its length to be equal to the contour's width along this axis.

## Mechanical model

The simulation model of puzzle shape emergence was adapted from Sapala et al (2018). The model is a fully damped mass-spring mechanical model written in C++ using the MorphoDynamX modeling framework (www.MorphoDynamX.org). In this model, cells are represented by point masses (vertices) linked together by linear springs. Based on the cell shape, additional intracellular connections (springs) are placed, representing the orientation of microtubules that guide the deposition of cellulose within the cell. Importantly, these connections are restricted to locations where geometric constraints, such as connection angle and curvature, permit their formation. The model can thus be used to predict the orientation and distribution of microtubules based on the outlines of cell walls obtained through microscopic imaging.

Growth is simulated by fixing the positions of the outermost vertices, moving them slightly outward at each simulation step before finding the positions of the inner vertices at the equilibrium between the forces on springs and the intracellular connection constraints. The rate at which the outer vertices are moved was varied to simulate distinct growth patterns. For uniform growth, the outer vertices move apart at a constant rate, possibly different in the horizontal and vertical directions, throughout the simulation. For nonuniform growth, the rate of movement of the outer vertices changes as the simulation runs.

The main changes to the model for this work were to add functions to allow for changing growth over time in the X and Y directions. These functions use the format of the "funcedit" function editing tool supplied with the VLab modeling environment (http://algorithmicbotany.org/virtual_laboratory/). This tool creates a function file (.func) that contains spline points to define the function. The function file can be edited interactively with funcedit or with a text editor. Step functions were created to simulate a period of full growth followed by a period of zero growth (StepDown.func) and vice versa (StepUp.func). Growth functions for the maize simulation were fit to growth data (ZeaGrowthX.func, ZeaGrowthY.func). A description of the model parameters, starting templates and growth processes can be found in the "description.txt" file contained in the model.

## Experimental design and statistical analysis

Sample sizes were chosen based on common practice in the field and feasibility; no statistical power analysis was performed. Samples were not randomized, and genotypes and treatments were processed in parallel where possible. Blinding was not performed during data collection or analysis. No samples were excluded from analysis, and no outlier removal was performed unless explicitly stated. Statistical tests and summary statistics are reported in the figure legends; exact $p$ values are provided where applicable. For model-based analyses, residuals were inspected to verify that model behavior was reasonable.

# Data availability

Experimental data, simulation models, additional MorphoDynamX processes, and the MorphoDynamX software are available online on Zenodo (https://doi.org/10.5281/zenodo.15184042). See the ReadMe.txt file for a description of the archived contents. The

following plant organ scans are modified from: *Amaranthus caudatus*: NEU000004163. *Neuchâtel Herbarium. Alstroemeria aurea*: BR0000025021905. Meise Botanic Garden Herbarium. *Fuchsia magellanica*: 304262. Field Museum of Natural History. *Mentha × piperita*: 00058444P. Oxford University Herbarium. *Cuphea ignea*: 289029. Oxford University Herbarium.

The source data of this paper are collected in the following database record: biostudies:S-SCDT-10_1038-S44319-026-00755-y.

## Peer review information

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

## Acknowledgements

We are grateful to Olivier Hamant, Arezki Boudaoud and Alice Malivert for comments on the manuscript. This work was supported by the University of Lausanne and the Swiss National Science Foundation (SNSF) Starting Grant PS00-3_234905 awarded to MM, the John Innes Foundation (NT), the EMBO Postdoctoral Fellowship (LH), NSF IOS 1553030 (AHKR), the National Science Centre, Poland (SHENG1 grant 2018/30/Q/NZ3/00189) (DK), ERA-CAPS awarded to DK and RSS, the Natural Sciences and Engineering Research Council of Canada Discovery Grant 2021-02795 (AR), and the Biotechnology and Biological Sciences Research Council (BBSRC) Institute Strategic Program Grant (BB/X01102X/1) to the John Innes Centre (RSS). We acknowledge Dr Izabela Potocka from the Scanning Electron Microscopy Laboratory of Institute of Biology, Biotechnology and Environmental Protection, University of Silesia for help in SEM of replicas. We thank Dr Azahara Martin for providing us with access to the phase-contrast microscope. We would like to thank Prof Geoffrey Wasteneys for providing *any1* seeds, and the Nottingham Arabidopsis Stock Center for distributing other seeds. The confocal microscopy was performed at the Cellular Imaging Facility (CIF), University of Lausanne.

## Author contributions

**Nicola Trozzi**: Data curation; Formal analysis; Funding acquisition; Investigation; Methodology; Writing—review and editing. **Brendan Lane**: Investigation; Writing—review and editing. **Alice Perruchoud**: Investigation. **Frances Clark**: Formal analysis; Investigation. **Lukas Hoermayer**: Investigation. **Andrea Meraviglia**: Investigation. **Tammo Reichgelt**: Investigation; Writing—review and editing. **Adrienne H K Roeder**: Formal analysis; Supervision; Funding acquisition; Writing—review and editing. **Dorota Kwiatkowska**: Formal analysis; Funding acquisition; Investigation; Writing—review and editing. **Adam Runions**: Formal analysis; Funding acquisition; Investigation; Writing—review and editing. **Richard S Smith**: Conceptualization; Data curation; Formal analysis; Supervision; Funding acquisition; Investigation; Methodology; Writing—review and editing. **Mateusz Majda**: Conceptualization; Data curation; Formal analysis; Supervision; Funding acquisition; Investigation; Methodology; Writing—original draft; Writing—review and editing.

Source data underlying figure panels in this paper may have individual authorship assigned. Where available, figure panel/source data authorship is listed in the following database record: biostudies:S-SCDT-10_1038-S44319-026-00755-y.

## Disclosure and competing interests statement

The authors declare no competing interests.

