## [Peer Review File · EMBO Reports]

Growth history leaves a geometric trace in puzzle cells

Nicola Trozzi, Brendan Lane, Alice Perruchoud, Frances Clark, Lukas Hoermayer, Andrea Meraviglia, Tammo Reichgelt, Adrienne Roeder, Dorota Kwiatkowska, Adam Runion, Richard Smith, and Mateusz Majda

Corresponding author(s): Mateusz Majda (mateusz.majda@unil.ch) , Richard Smith (Richard.Smith@jic.ac.uk)

Review Timeline:	Transfer Date:	15th Sep 25
	Editorial Decision:	9th Dec 25
	Revision Received:	4th Jan 26
	Editorial Decision:	23rd Feb 26
	Revision Received:	27th Feb 26
	Accepted:	16th Mar 26

Transaction Report: This manuscript was transferred to EMBO reports following peer review at The EMBO Journal.

Referee #1:

In this interesting manuscript Trozzi and colleagues undertake an extensive and multifaceted investigation into the phenomenon of plant epidermal cell lobing. The authors start from the premiss explored by Sapala and colleagues that lobing is an outcome of cellular responses to geometry-dependent turgor-driven mechanical stresses which change dynamically during cell expansion. They extend existing parameters previously used to quantitatively analyse lobing by Sapala and colleagues. They also extend the functionality of the innovative mechanistic model previously developed by Sapala and colleagues to allow the introduction of variable growth directionality (isotropy/anisotropy) over time. This modified model is used here both to explore the influence of growth history on cell lobing, and the potential influence of altering the cellular responses that lead to cell lobing on tissue/organ growth. Predictions are tested using a number of systems including developing maize leaves and a range of Arabidopsis mutants, as well as Arabidopsis plants subjected to pharmacological treatments. The authors then go on to survey the occurrence and characteristics of lobing across both extinct and extant land plants. Their observation that although most plants seem to be able to produce lobed cells, this capacity is not necessarily expressed in all tissues/organs, and is also environmentally sensitive, is highly intriguing and supports the hypothesis that multiple factors, potentially both molecular and mechanical/geometric, could be at play.

Because of its important role in underpinning this current work, before writing this review, I reread the Sapala et al., paper. Unfortunately, after doing this, I was left in some doubt as to whether the work in the current study truly provides the type of conceptual advance that I initially thought it might. Instead, I feel that this work, although solid and strongly confirmatory of the Sapala study, is in essence rather descriptive and presents conceptual advances that are somewhat incremental. The main novelty lies in the "growth history analysis". But I'm not sure that the findings are enough to push this work over the line.

Could the prediction that cell shape might be used to reconstruct growth history be tested more directly, for example using transient pharmacological treatments?

The impact of the current work could be enhanced by more thoroughly addressing one (or more) of the following areas:

- 1) The influence of the timing of cell division arrest on lobing. The authors introduce the influence of altering the timing of cell division arrest on lobing quite late in the manuscript. Their discussion of the potential role of this factor (which is intuitively quite obvious) is very thorough and pertinent, and observations in diverse species and some Arabidopsis mutants support this role. However, the impact of cell division is not investigated using the mechanistic model. Cell division arrest is clearly a critical factor in determining final cell size and lobing, but division orientation would be influenced by growth anisotropy, altering

cell geometries and mechanics very dynamically. Modelling this could well reveal some less intuitive outcomes. Although I realise that adding this aspect to the model is likely a complex undertaking, I feel it would really take this study to the next level, and build stronger bridges with the ongoing work of several other teams.

2) The influence of the third dimension on the phenomena described. Epidermal cells are not two dimensional objects, and this factor is not really taken into account either in the Sapala paper or in this work. The third dimension (cell depth and curvature of the external cell wall) could significantly influence the magnitude of the stress inferred from the "Largest Empty Circle" parameter, and likely evolves differently to surface parameters during growth. Although I again realise that adding a third dimension to the mechanistic model could be complex, the fact that this factor is not even discussed is a little frustrating.

3) The influence of subepidermal tissue on stress and lobing. Neighbouring cells, both within the epidermis (contributing to tissue-level tension coordinating growth, the phenomenon described in this work) but also underlying the epidermis (again contributing to tissue/organ level tension but also potentially physically constraining epidermal cells) will influence the 3D shape (see point 2) of any given epidermal cell. Addressing this aspect is clearly one of the key frontiers in this research field, and could have been fruitfully initiated in the current work. This point is, for example, of importance when considering differences in lobing at different points on the same leaf. For example, in figure S7 (line 258: species with enhanced lobing on the abaxial leaf surface) the authors seem to suggest that differential exposure to light might influence lobing, but it is entirely possible that differences in tissue level tension (due to the concave nature of the abaxial surface) or differences in interactions with mesophyll cells, could be equally important. This difference in adaxial vs abaxial lobing is also seen in Arabidopsis, potentially facilitating further exploration.

More minor points

The current manuscript is well written and pleasant to read, and is well illustrated.

Occasionally too much information is missing from main figures to allow them to "stand alone". This is particularly true for Figure 1. Some information from Figure S1 should be included in this figure to allow readers to rapidly understand the dynamics of growth that have been applied to give the outcomes shown.

Figure 3: The X axis on graphs should state what is being analysed.

Referee #2:

Line 51-52: There are Arabidopsis mutants that have less lobes. Is there any evidence that

cell integrity in these mutants is disrupted and they explode? or are damaged in some way? if not, this explanation is more speculative and i would amend it.

line 59: growth may rely on turgor and mechanical properties but whether coordination of growth between cells is dependent on that is highly questionable! that would almost require constant dynamic change in in these two which I am not anyone has shown.

Line 89: While the notion that final cell shape "records" the history of tissue growth is conceptually appealing, it risks reversing the causal direction. Morphogenesis is not driven by past growth, but by present, dynamically evolving growth patterns. Rather than viewing shape as a passive archive of growth history, it may be more accurate to see it as the continuously updated outcome of local, spatiotemporally resolved growth cues.

Emphasizing "growth history" as the determinant of final shape can inadvertently obscure the active, moment-to-moment integration of mechanical and developmental signals that drive cell morphogenesis. If you wish to present shape outcome as "growth history", you should clearly lay out that this idea is the continuous sequence of organ-scale growth patterns that dynamically influence cell shape in each present moment. It could for instance be stated as: "Cell shape reflects the cumulative outcome of dynamically evolving growth patterns, with changes in local growth anisotropy influencing morphogenetic processes in real time."

line 103: It is important to note that this is only model. All one can say is that it suggest a plausible explanation not really a mechanistic explanation, that would require demonstration of this!

Line 110: The phrase "temporal changes in growth anisotropy" (line 116-117) is somewhat ambiguous and might be interpreted as general fluctuation over time. However, the point being demonstrated in the modeling is more specific - that the sequential order of anisotropic and isotropic growth phases significantly impacts final cell shape, even when cumulative growth is held constant. To improve clarity, I suggest rephrasing this sentence to something like: "We further explored how the sequence of anisotropic and isotropic growth phases affects cell morphology, even when the total cumulative growth is the same."

Line 116: To make this sentence clearer: You are referring to the sequential

Line 122. The statement that the simulated cells "closely resemble cells observed in some areas of the leaf in Arabidopsis" (lines 121-122) is vague and currently unsubstantiated. It would strengthen the manuscript to specify which part of the leaf this refers to (e.g., proximal vs. distal, adaxial vs. abaxial), ideally with corresponding micrographs or segmentation overlays for comparison. Without this, the resemblance remains subjective.

Supplemental figure 2: This figure is a central piece of data for this manuscript, but is hard to follow. Could you please make some kind of overview image to more clearly convey what

regions are being tracked and compared. You also make claims about strain rates here, from SEM images. Supposedly, this is non-growing (dead) material, suggesting that you are not following actual growth. Unless I have misunderstood how this data was generated, I fear that this growth tracking is highly subjective since you are manually selecting regions to examine at different timepoints. I would expect that it is completely plausible to perform real live time-lapse imaging series on maize leaves, either stained with propidium iodide, marking leaf regions with an ink marker to identify regions over time. Can the authors please clarify how this data represents a reliable time-lapse dataset that accurately captures growth over time in maize leaves (with accompanying clarification in the figure)? Alternatively, the authors should attempt to perform a true time-lapse imaging of maize leaves

line 141: of leaf? or of the cells?

Line 152-153. The statement that your work "reveals a novel dynamic feedback mechanism in which organ-scale growth dynamics significantly influence individual cell morphology" (lines 152-154) overstates the novelty of this concept. The influence of tissue-scale growth dynamics on epidermal cell shape has been reported previously in both experimental and modeling studies. Rather than presenting this relationship as novel, I suggest rephrasing to emphasize what is new in your approach - for example, your demonstration that the sequence of anisotropic and isotropic growth phases can lead to distinguishable shape outcomes, or your large-scale comparative analysis across taxa and organs. Reframing the novelty in this direction would make the manuscript more accurate and impactful.

Line 171: Is there a reason for abandoning the LEC concept? In other words, did the authors compare the capacity of LEC versus Min-axis to reveal correlation with lobeyness?

Line 182: What leads you to primarily suggest that all of these proposed compensatory mechanisms respond specifically to the absence of puzzle-like lobes? This link seems highly correlative and I don't see the necessity for a causal arrow between the two.

line 188: I think authors need to measure growth of leaf cells, both size and shape and compare it with wild type and not rely on overall stature to reach conclusions.

Line 196: The interpretation that altered cell shapes (e.g., elongation or zigzag morphology) in mutants like *ctr1* or *any1* are compensatory mechanisms in response to lobe loss (lines 195-197) seems speculative. These changes could equally result from altered cell wall biosynthesis, growth anisotropy, or signaling disruptions inherent to the mutant backgrounds, rather than reflecting an active mechanical compensation strategy. Without functional evidence showing that these changes restore wall stress distribution or actively compensate for mechanical instability, I suggest rephrasing this claim more cautiously. For example: "...these mutants exhibit altered cell shapes that may reflect changes in growth behavior associated with disrupted lobe formation, although whether these represent true compensatory mechanisms remains unclear."

Line 199: Again, this statement conflates causation with correlation. The authors must be more careful with overinterpreting their data.

line 209: This is tricky: a consequence of reduced endoreduplication is entry into divisions and when cells divide more and expand less, they are smaller in size i. e. due to faster divisions, thus there could be less time to expand so the consequence on lobeyness could be indirect. this should be discussed!

line 214: I dont understand why the authors switch from maize to Arabidopsis?

loine 226: The issue i would like to raise is that everything that is used to disrupt cell shape is also an indirect or direct modifier of cell wall or growth so it is difficult to disentangle easily these relationships. At least these limitations should be discussed

line 302: would this be moderate? seems low to me

line 341-342: It is important to note that demonstrating that puzzle cell shape helps to reduce stress may be true but actually proving it is rather difficult. correlations are not really proof. This must be discussed or rephrased.

line 372: Indeed this is what one would expect

line 375: really this is point: lobing is a temporal event and if cells are smaller and dont get enough time to expand, they will form less lobes as there is not sufficient time. I mean even in wild type, lobes develop over time. so early on there are fewer lobes.

line 385: I think it is response but i am not sure it is strategic one! for example, if a mutant is disrupted in cold acclimation, it will die, so development of cold acclimation is a strategic response or adaptation.

Referee #1:

In this interesting manuscript Trozzi and colleagues undertake an extensive and multifaceted investigation into the phenomenon of plant epidermal cell lobing. The authors start from the premiss explored by Sapala and colleagues that lobing is an outcome of cellular responses to geometry-dependent turgor-driven mechanical stresses which change dynamically during cell expansion. They extend existing parameters previously used to quantitatively analyse lobing by Sapala and colleagues. They also extend the functionality of the innovative mechanistic model previously developed by Sapala and colleagues to allow the introduction of variable growth directionality (isotropy/anisotropy) over time. This modified model is used here both to explore the influence of growth history on cell lobing, and the potential influence of altering the cellular responses that lead to cell lobing on tissue/organ growth. Predictions are tested using a number of systems including developing maize leaves and a range of *Arabidopsis* mutants, as well as *Arabidopsis* plants subjected to pharmacological treatments. The authors then go on to survey the occurrence and characteristics of lobing across both extinct and extant land plants. Their observation that although most plants seem to be able to produce lobed cells, this capacity is not necessarily expressed in all tissues/organs, and is also environmentally sensitive, is highly intriguing and supports the hypothesis that multiple factors, potentially both molecular and mechanical/geometric, could be at play.

Because of its important role in underpinning this current work, before writing this review, I reread the Sapala et al., paper. Unfortunately, after doing this, I was left in some doubt as to whether the work in the current study truly provides the type of conceptual advance that I initially thought it might. Instead, I feel that this work, although solid and strongly confirmatory of the Sapala study, is in essence rather descriptive and presents conceptual advances that are somewhat incremental. The main novelty lies in the "growth history analysis". But I'm not sure that the findings are enough to push this work over the line. Could the prediction that cell shape might be used to reconstruct growth history be tested more directly, for example using transient pharmacological treatments?

The impact of the current work could be enhanced by more thoroughly addressing one (or more) of the following areas:

Reply: As requested, we have rewritten the text to make the novelty and focus clear, and tested the hypothesis with biological data. We have replied to this point below.

1) The influence of the timing of cell division arrest on lobing. The authors introduce the influence of altering the timing of cell division arrest on lobing quite late in the manuscript. Their discussion of the potential role of this factor (which is intuitively quite obvious) is very thorough and pertinent, and observations in diverse species and some *Arabidopsis* mutants support this role. However, the impact of cell division is not investigated using the mechanistic model. Cell division arrest is clearly a critical factor in determining final cell size and lobing, but division orientation would be influenced by growth anisotropy, altering cell

geometries and mechanics very dynamically. Modelling this could well reveal some less intuitive outcomes. Although I realise that adding this aspect to the model is likely a complex undertaking, I feel it would really take this study to the next level, and build stronger bridges with the ongoing work of several other teams.

Reply: We have addressed this by extending the model to include cell division, as shown in the updated Figure 1 (G-I) and new Suppl. Movies 4-6. These panels replicate the growth scenarios in (D-F) but incorporate divisions that cease after the first 20 of 100 growth steps, revealing how even a relatively short delay in division arrest substantially alters lobe formation and direction. The corresponding section in the Results (lines 157-168) has been rewritten to describe these new simulations.

2) The influence of the third dimension on the phenomena described. Epidermal cells are not two dimensional objects, and this factor is not really taken into account either in the Sapala paper or in this work. The third dimension (cell depth and curvature of the external cell wall) could significantly influence the magnitude of the stress inferred from the "Largest Empty Circle" parameter, and likely evolves differently to surface parameters during growth. Although I again realise that adding a third dimension to the mechanistic model could be complex, the fact that this factor is not even discussed is a little frustrating.

Reply: We have expanded our discussion to emphasize the three-dimensional organization of pavement cells and the rationale behind the LEC approximation at lines 54-60. Previous 3D finite-element modeling (10.7554/eLife.32794) has shown that the LEC metric captures the maximal unsupported span of the cell wall under turgor pressure, rather than just its two-dimensional projection. At the stage when puzzle cells first emerge, differences in curvature between the abaxial and adaxial leaf surfaces are minor compared with the local curvature of individual cells. The substantial curvature of the outer periclinal wall under pressure is directly incorporated into stress calculations. Accordingly, our use of the LEC metric, supported by prior 3D FEM simulations, provides a robust approximation of the 3D stresses experienced by the cells.

3) The influence of subepidermal tissue on stress and lobing. Neighbouring cells, both within the epidermis (contributing to tissue-level tension coordinating growth, the phenomenon described in this work) but also underlying the epidermis (again contributing to tissue/organ level tension but also potentially physically constraining epidermal cells) will influence the 3D shape (see point 2) of any given epidermal cell. Addressing this aspect is clearly one of the key frontiers in this research field, and could have been fruitfully initiated in the current work. This point is, for example, of importance when considering differences in lobing at different points on the same leaf. For example, in figure S7 (line 258: species with enhanced lobing on the abaxial leaf surface) the authors seem to suggest that differential exposure to light might influence lobing, but it is entirely possible that differences in tissue level tension (due to the concave nature of the abaxial surface) or differences in interactions with

mesophyll cells, could be equally important. This difference in adaxial vs abaxial lobing is also seen in Arabidopsis, potentially facilitating further exploration.

Reply: The influence of subepidermal tissues on epidermal lobing is indeed an important frontier. To our knowledge, there is currently very little experimental study linking puzzle cell formation to coordinated growth between the epidermis and underlying layers. The mesophyll contains air spaces, and it has been suggested that, at this developmental stage, planar growth is driven primarily by the epidermis rather than by internal tissues (doi.org/10.1371/journal.pbio.2005952). Once those air spaces form, however, the internal cell walls bordering them will experience considerable unbalanced stress, since they lack pressure support from adjacent cells. In that context, the LEC concept applies, cells should minimize large unsupported areas. Indeed, there is evidence that, as mesophyll air spaces enlarge, the surrounding cells adopt a three-dimensional, puzzle-like shape (doi.org/10.1186/1746-4811-6-17). We have now expanded the Discussion (lines 500-506) to clarify that adaxial–abaxial differences in lobing could arise from multiple factors, including light exposure, tissue curvature, and interactions with mesophyll cells. A mechanistic treatment of these interactions lies beyond the scope of the present work and would require a separate study

More minor points

The current manuscript is well written and pleasant to read, and is well illustrated. Occasionally too much information is missing from main figures to allow them to "stand alone". This is particularly true for Figure 1. Some information from Figure S1 should be included in this figure to allow readers to rapidly understand the dynamics of growth that have been applied to give the outcomes shown.

Reply: We have expanded Figure 1 to include simulations that incorporate cell divisions. We have also moved the maize-related simulation and time-lapse imaging data from the original Figure 1 and Supplementary Figure 1 into a new main Figure 2. This restructuring allows readers to directly connect model predictions with observed growth dynamics in maize, making both the modeling and its biological relevance clearer.

Figure 3: The X axis on graphs should state what is being analysed.

Reply: We have edited Figure 3 to include a clear indication of the analysis being done on the X axis of the graphs. We have also improved the leaf scaling by adding mutant leaves with the same scale as the WT to show the difference in size.

Referee #2:

Line 51-52: There are Arabidopsis mutants that have less lobes. Is there any evidence that cell integrity in these mutants is disrupted and they explode? or are damaged in some way? if not, this explanation is more speculative and i would amend it.

Reply: The original phrasing may have overstated the functional consequences of lobe loss. While lobes reduce wall stress during growth, current evidence does not show that mutants with reduced lobing experience overt cell damage or loss of integrity.

To address this, we revised the text at lines 51–52 (now 67-72) from “*Finite element modeling and stress analysis on puzzle cell shapes provides a functional explanation for these unusually shaped cells, demonstrating that lobed shapes significantly lower mechanical stress during more isotropic expansion.*” to “*Finite element modeling and 3D stress analysis support a functional role for these unusual shapes, showing that lobed cells experience reduced wall stress during isotropic expansion. This suggests that puzzle shape formation reflects a mechanical adaptation to accommodate endoreduplication-induced cell enlargement and organ scale growth dynamics, positioning mechanical stress as a key constraint in shaping epidermal cell morphology*”

line 59: growth may rely on turgor and mechanical properties but whether coordination of growth between cells is dependent on that is highly questionable! that would almost require constant dynamic change in in these two which I am not anyone has shown.

Reply: The extent to which dynamic changes in turgor and wall mechanics mediate intercellular coordination is still under investigation. Our intention was not to claim that coordination is strictly dependent on mechanical factors, but rather to reflect the idea that these properties constrain how cells grow relative to one another. Several studies support the idea that mechanical stress patterns can propagate across tissues and influence growth direction or cytoskeletal alignment in neighboring cells (e.g., [10.1126/science.1165594](https://doi.org/10.1126/science.1165594) , [10.7554/eLife.01967](https://doi.org/10.7554/eLife.01967) , [10.1016/j.cell.2012.02.048](https://doi.org/10.1016/j.cell.2012.02.048)). To clarify this point and avoid overstatement, we have revised that sentence from: “*This coordination relies on the interplay between internal turgor pressure and the mechanical properties of the cell wall.*” to: “*This coordination is influenced by the interplay between internal turgor pressure and the mechanical properties of the cell wall, which together constrain how neighboring cells can grow relative to each other.*”

Line 89: While the notion that final cell shape "records" the history of tissue growth is conceptually appealing, it risks reversing the causal direction. Morphogenesis is not driven by past growth, but by present, dynamically evolving growth patterns. Rather than viewing shape as a passive archive of growth history, it may be more accurate to see it as the continuously updated outcome of local, spatiotemporally resolved growth cues. Emphasizing "growth history" as the determinant of final shape can inadvertently obscure the active, moment-to-moment integration of mechanical and developmental signals that drive cell morphogenesis. If you wish to present shape outcome as "growth history", you should clearly lay out that this idea is the continuous sequence of organ-scale growth patterns that dynamically influence cell shape in each present moment. It could for instance be stated as:

"Cell shape reflects the cumulative outcome of dynamically evolving growth patterns, with changes in local growth anisotropy influencing morphogenetic processes in real time."

Reply: Cell shape emerges from the dynamic integration of growth cues rather than acting as a passive record of past growth. Our intention was to convey that temporally varying growth anisotropy can leave geometric signatures that persist in final cell shape. To clarify this, we revised the sentence to: *"These observations suggest that final cell shape reflects the cumulative outcome of dynamically evolving growth patterns, with changes in local growth anisotropy influencing morphogenesis in real time."*

line 103: It is important to note that this is only model. All one can say is that it suggest a plausible explanation not really a mechanistic explanation, that would require demonstration of this!

Reply: We agree that the model offers a plausible scenario rather than a demonstrated mechanism. To avoid overstating its implications, we have revised the sentence to *"These simulations suggest that differences in the temporal dynamics of growth anisotropy, even under conditions of identical cumulative growth, can plausibly lead to significant variations in cell shape."*

Line 110: The phrase "temporal changes in growth anisotropy" (line 116-117) is somewhat ambiguous and might be interpreted as general fluctuation over time. However, the point being demonstrated in the modeling is more specific - that the sequential order of anisotropic and isotropic growth phases significantly impacts final cell shape, even when cumulative growth is held constant. To improve clarity, I suggest rephrasing this sentence to something like: "We further explored how the sequence of anisotropic and isotropic growth phases affects cell morphology, even when the total cumulative growth is the same."

Reply: The original phrasing could be misinterpreted as referring to random fluctuations rather than defined sequences of growth phases. To clarify this, we have revised the sentence to *"Building upon this finding, we further explored how the sequence of anisotropic and isotropic growth phases affects cell morphology, even when the total cumulative growth is the same."*

Line 116: To make this sentence clearer: You are referring to the sequential

Reply: We have clarified that we are referring to the specific sequence of anisotropic followed by isotropic growth (and *vice versa*), rather than general temporal variation. This change makes the modeling result more explicit. *"Conversely, reversing the growth sequence, starting with anisotropic growth followed by isotropic growth, resulted in horizontally biased lobes."*

Line 122. The statement that the simulated cells "closely resemble cells observed in some areas of the leaf in Arabidopsis" (lines 121-122) is vague and currently unsubstantiated. It

would strengthen the manuscript to specify which part of the leaf this refers to (e.g., proximal vs. distal, adaxial vs. abaxial), ideally with corresponding micrographs or segmentation overlays for comparison. Without this, the resemblance remains subjective.

Reply: The original phrasing was too vague. To address this, we now specify that the simulated cell shapes resemble those found in the proximal and distal regions on the abaxial side of the Arabidopsis leaf, where cells are larger and exhibit more lobing. To support this, we have added a new supplementary figure (Suppl. Figure 2) showing lobeyness heatmaps of confocal images of epidermal cells from proximal and distal regions of both adaxial and abaxial surfaces. These images illustrate the spatial variation in cell shape and provide a visual basis for comparison with the modeled geometries.

Supplemental figure 2: This figure is a central piece of data for this manuscript, but is hard to follow. Could you please make some kind of overview image to more clearly convey what regions are being tracked and compared. You also make claims about strain rates here, from SEM images. Supposedly, this is non-growing (dead) material, suggesting that you are not following actual growth. Unless I have misunderstood how this data was generated, I fear that this growth tracking is highly subjective since you are manually selecting regions to examine at different timepoints. I would expect that it is completely plausible to perform real live time-lapse imaging series on maize leaves, either stained with propidium iodide, marking leaf regions with an ink marker to identify regions over time. Can the authors please clarify how this data represents a reliable time-lapse dataset that accurately captures growth over time in maize leaves (with accompanying clarification in the figure)? Alternatively, the authors should attempt to perform a true time-lapse imaging of maize leaves

Reply: We are sorry for the lack of clarity. The images represent the time lapse of the growth tracking of the same cells over time (the method was previously used in: [10.1071/FP08047](https://doi.org/10.1071/FP08047); [10.1046/j.1365-3113X.2001.01350.x](https://doi.org/10.1046/j.1365-3113X.2001.01350.x); [10.1007/978-1-62703-643-6_8](https://doi.org/10.1007/978-1-62703-643-6_8); [10.1038/s41477-023-01452-7](https://doi.org/10.1038/s41477-023-01452-7); [10.1111/pce.12809](https://doi.org/10.1111/pce.12809)). We have now clarified in the text that the replicas were obtained from living tissue using *in vivo* impressions.. Because of its importance, we have reorganized the maize time-lapse data into a new main Figure 2, with supporting details in new Supplemental Figure 3.

line 141: of leaf? or of the cells?

Reply: We confirm that the sentence describes anisotropic epidermal cell growth within the growing maize leaf, oriented along the organ axis. To avoid confusion, we revised the sentence to clarify that it refers to the direction and dynamics of cell expansion, not to overall organ growth. “*Our observations revealed that puzzle-like shapes in maize pavement cells emerge from a two-phase growth pattern: an initial phase of strong anisotropic expansion along the leaf’s longitudinal axis, followed by a shift to nearly isotropic growth, during which transverse expansion increases relative to longitudinal expansion.*”

Line 152-153. The statement that your work "reveals a novel dynamic feedback mechanism in which organ-scale growth dynamics significantly influence individual cell morphology" (lines 152-154) overstates the novelty of this concept. The influence of tissue-scale growth dynamics on epidermal cell shape has been reported previously in both experimental and modeling studies. Rather than presenting this relationship as novel, I suggest rephrasing to emphasize what is new in your approach - for example, your demonstration that the sequence of anisotropic and isotropic growth phases can lead to distinguishable shape outcomes, or your large-scale comparative analysis across taxa and organs. Reframing the novelty in this direction would make the manuscript more accurate and impactful.

Reply: Prior studies have described how tissue-scale growth patterns influence cell morphology. Our intention was to highlight the novel aspects of our approach, particularly the demonstration that the sequence of anisotropic and isotropic growth phases can lead to distinct cell shapes, as well as the breadth of our comparative analysis across organs and species. We have revised the sentence to more accurately reflect the contributions of our study "*By integrating empirical observations with computational modeling, we show that the sequence of anisotropic and isotropic growth phases can explain the puzzle cell shapes observed in maize, demonstrating how temporal growth dynamics shape cell morphology in planta.*"

Line 171: Is there a reason for abandoning the LEC concept? In other words, did the authors compare the capacity of LEC versus Min-axis to reveal correlation with lobeyness?

Reply: In Figure 3 (previously Figure 2), our goal was to separate two situations: cells constrained in two directions, which later develop lobes, from elongated cells, where lobes are unnecessary. For this purpose, we used min-axis. This parameter measures the smallest cell diameter and therefore reports growth constraints before any lobes appear. By design, it is unaffected by lobing itself. In contrast, LEC changes as lobes form and thus reflects the outcome of shape elaboration rather than the underlying constraint. For this reason, we used min-axis for the population-level analysis, while keeping LEC as the basis for the mechanical interpretation presented earlier in the manuscript.

Line 182: What leads you to primarily suggest that all of these proposed compensatory mechanisms respond specifically to the absence of puzzle-like lobes? This link seems highly correlative and I don't see the necessity for a causal arrow between the two.

Reply: Our original phrasing implied a stronger causal relationship than is warranted by the available data. Our intention was to describe associated phenotypes that may reflect a developmental or mechanical adjustment in the context of reduced lobeyness, rather than to assert that these mechanisms are directly triggered by the absence of lobes. We have revised the sentence from "*This suggests that these cells may either experience lower mechanical stress, possibly due to changes in the arrangement of neighboring cells and tissue structure,*

or they might activate compensatory mechanisms, such as restricted organ growth, increased cell elongation or altered cell wall properties, in response to the absence of puzzle-like lobes.” to “ One possible explanation is that these cells experience lower mechanical stress, possibly due to changes in the arrangement of neighboring cells and tissue structure. Another hypothesis is that they have compensatory traits such as restricted organ growth, increased cell elongation, or altered cell wall properties.” to reflect this.

line 188: I think authors need to measure growth of leaf cells, both size and shape and compare it with wild type and not rely on overall stature to reach conclusions.

Reply: In response, we have added a completely new Figure 4 that presents live time-lapse measurements of cell-level growth, shape (area and lobeyness), lateral versus longitudinal expansion, and division frequency in WT Col-0 versus the miR319-overexpressing *jaw-D* mutant from 6–8 days after sowing (DAS). By quantifying width-to-length expansion ratios (Fig. 4A–B), cell areas (Fig. 4C–D), lobeyness (Fig. 4E–F), and division heatmaps (Fig. 4J–K), and summarizing these metrics in violin plots (Fig. 4G–I), width/length ratios (Fig. 4L) and total cell counts (Fig. 4M), we now directly compare how *jaw-D* alters cellular behaviors relative to Col-0. These analyses link the broader, more triangular leaf form of *jaw-D* with specific changes in cell growth orientation, size, shape, and proliferation, rather than relying solely on whole-leaf stature.

Line 196: The interpretation that altered cell shapes (e.g., elongation or zigzag morphology) in mutants like *ctrl* or *any1* are compensatory mechanisms in response to lobe loss (lines 195-197) seems speculative. These changes could equally result from altered cell wall biosynthesis, growth anisotropy, or signaling disruptions inherent to the mutant backgrounds, rather than reflecting an active mechanical compensation strategy. Without functional evidence showing that these changes restore wall stress distribution or actively compensate for mechanical instability, I suggest rephrasing this claim more cautiously. For example: “...these mutants exhibit altered cell shapes that may reflect changes in growth behavior associated with disrupted lobe formation, although whether these represent true compensatory mechanisms remains unclear.”

Reply: The original wording implied a stronger causal relationship than our data can support. Our intention was to describe the observed association between reduced lobeyness and altered cell shapes in various mutants, while acknowledging that these changes could arise from diverse underlying causes. We have revised the text to present this interpretation more cautiously and clarify that the compensatory nature of these phenotypes remains speculative in the absence of functional evidence: “*Collectively, these mutants indicate that reduced lobeyness is associated with altered cell shapes, such as elongation or zigzag morphologies. These changes could reflect responses to growth constraints but may also result from underlying disruptions in cell wall synthesis, anisotropy, or signaling pathways specific to*

each mutant background. Without direct functional evidence, the extent to which these altered shapes represent compensatory mechanisms remains unclear.”

Line 199: Again, this statement conflates causation with correlation. The authors must be more careful with overinterpreting their data.

Reply: We have revised the sentence to avoid implying a causal relationship between lobeyness and overall plant growth. The updated text now emphasizes the observed correlation without suggesting direct causality: *“These observations reveal a consistent association between reduced lobeyness, smaller epidermal cells, and reduced plant growth. However, whether the loss of lobes contributes causally to these growth defects remains to be tested.”*

line 209: This is tricky: a consequence of reduced endoreduplication is entry into divisions and when cells divide more and expand less, they are smaller in size i. e. due to faster divisions, thus there could be less time to expand so the consequence on lobeyness could be indirect. this should be discussed!

Reply: The effect of endoreduplication on lobeyness is likely indirect. To clarify this, we have revised the text: *“These observations are consistent with the idea that extended proliferation limits the expansion phase, restricting lobe development, and that changes in lobeyness arise indirectly from altered growth dynamics downstream of endoreduplication”*

line 214: I dont understand why the authors switch from maize to Arabidopsis?

Reply: We clarified in the text that maize was used to study sequential tissue-scale growth anisotropy, while Arabidopsis provided a tractable genetic and pharmacological system to test lobing effects independently of long-range growth. First, we added some explanation for moving from Arabidopsis models to maize models and growth studies *“While these Arabidopsis models reveal how cell division timing influences lobing under different growth patterns, we next asked whether similar principles apply in a species with strong directional growth. [...] We focused on maize (Zea mays), whose elongated leaves contain pavement cells with lobes oriented mainly in the transverse direction, an arrangement that is unexpected given the leaf shape, which suggests growth anisotropy should favor the longitudinal axis.”* and then back from maize to Arabidopsis: *“While maize was used to study how temporal changes in tissue-level anisotropy influence cell shape, Arabidopsis cotyledons expand nearly isotropically and are amenable to dose- and time-controlled perturbations of cytoskeletal components.”*

line 226: The issue i would like to raise is that everything that is used to disrupt cell shape is also an indirect or direct modifier of cell wall or growth so it is difficult to disentangle easily these relationships. At least these limitations should be discussed

Reply: The perturbations used to reduce lobing, whether genetic or pharmacological, also influence other aspects of growth and cell wall dynamics. Disentangling the specific contribution of cell shape from these broader effects is inherently difficult. However, we observed consistent phenotypes across independent perturbations, including cytoskeletal drugs, endoreduplication mutants, and cellulose-deficient lines, all of which reduce lobeyness and result in smaller organs. This convergence suggests that altered lobing itself contributes to growth defects, even if not in isolation. We have added a sentence to the manuscript discussing this limitation: *“Although the perturbations used here broadly affect growth and cell wall dynamics in different ways, the consistency of phenotypic outcomes across independent manipulations supports the idea that reduced lobing contributes to the observed defects.”*

line 302: would this be moderate? seems low to me

Reply: We have changed *“moderate or higher positive correlation”* to *“significant positive correlation”*, which is more accurate.

line 341-342: It is important to note that demonstrating that puzzle cell shape helps to reduce stress may be true but actually proving it is rather difficult. correlations are not really proof. This must be discussed or rephrased.

Reply: Correlation does not establish causation, and demonstrating a mechanistic role for puzzle cell shapes in reducing mechanical stress would require further experimental validation. Our current data support an association between cell shape and stress patterns, but we do not claim this as definitive proof. We have rephrased the text to reflect this distinction: *“This finding reveals a consistent association between increased lobing and reduced LEC expansion relative to growth, supporting the idea that lobing may help to limit wall free span and reduce tensile stress. However, whether these shapes actively buffer stress or simply arise in response to mechanical constraints remains to be tested.”*

line 375: really this is point: lobing is a temporal event and if cells are smaller and dont get enough time to expand, they will form less lobes as there is not sufficient time. I mean even in wild type, lobes develop over time. so early on there are fewer lobes.

Reply: Lobing is a time-dependent process and smaller or rapidly dividing cells may exhibit reduced lobeyness simply because they have had less time to form lobes. This interpretation aligns with previous observations in wild-type tissues, where lobe formation progresses during the course of cell expansion. To clarify this point, we have added a sentence to the text highlighting the temporal nature of lobing and its implications for interpreting reduced lobeyness in mutants or treatments affecting cell size or division rate: *“However, puzzle cell shape is only required when cells become large in more than one direction, and multiple strategies may mitigate this need. For example, some species, such as yellow guava (Psidium guajava), appear to maintain smaller, more isodiametric epidermal cells, potentially through*

higher division rates, which would limit cell enlargement and thus lessen the mechanical demand for lobing.”

line 385: I think it is response but i am not sure it is strategic one! for example, if a mutant is disrupted in cold acclimation, it will die, so development of cold acclimation is a strategic response or adaptation.

Reply: The term “strategic” may overstate the interpretation and we have revised the sentence to refer more generally to puzzle cell formation as a developmental response to mechanical stress: “Collectively, these observations support the view that puzzle cell formation emerges as a developmental response to mitigate the high mechanical stresses that would be created in large isodiametric cells, and that it is mediated by growth restriction, a process that is likely broadly conserved in vascular plants.”

Dear Mateusz,

Thank you for the submission of your revised manuscript to EMBO reports. I sincerely apologize for the delay in handling your manuscript, but we have now received the full set of referee reports that is copied below.

As you will see, both referees are very positive about the study and recommend publication. Before I can formally accept the manuscript, I need you to format your manuscript according to our guidelines. Once you have submitted the revised version, we will perform a number of quality control steps, including a routine figure check. In order to speed up this process, I note below a few things that I noticed and that will need to be addressed, as well as a few points that are often overlooked, in addition to the more general formatting information further down.

I will also send you a separate e-mail with instructions how to provide source data for your manuscript.

=====

SPECIFIC POINTS:

- We need individual production quality figure files.
- Supplementary Figures should be combined into one PDF, called Appendix. The nomenclature is Appendix Fig. S#. The Appendix needs a table of content with page numbers. Alternatively, you can promote up to five figures to the "Expanded View". Please see more information below.
- Supplementary movies: the correct nomenclature is Movie EV#. Their legend needs to be provided as a README.txt file and then the legend and the movie are zipped together, and the zip file (one per movie) is uploaded.
- Please remove the author contributions from the manuscript file and make sure that the information in the online submission system is up to date and accurate. The information on author contributions specified in the manuscript tracking system will be typeset into the article.
- Make sure the information on funding in the manuscript matches that specified in the online manuscript submission system. The information must be congruent.
- Scale bars must be defined in the figure legend only, not in the figure itself.
- Figure 4 L, M, Figure 5 R, S: please show the individual datapoints in addition to the mean (?) and error bars.
- Figure 6J: please define the number of samples/experiments and their nature (technical, biological).
- Supplementary Table 1 should be Table EV1. Please also update the name in the .xls file itself.
- We need a Disclosure and competing interests statement .
- The reference after the legend of Figure 7 appears out of place (Harline K et al, 2022).
- There are more references after the data availability section/methods. These should be part of the general reference section.
- The manuscript sections should be in this order:
Title page - Abstract - Introduction - Results - Discussion - Methods - Acknowledgements - Disclosure and competing interests statement - References - Figure legends - Tables and their legends (not EV tables) - Expanded View Figure legends
- We need 5 keywords on the title page.
- We will need an Author Checklist and a Reagents and Tools table (see below for more information).

=====

GENERAL formatting guidelines:

2) individual production quality figure files as .eps, .tif, .jpg (one file per figure).

Please download our Figure Preparation Guidelines (figure preparation pdf) from our Author Guidelines pages <https://www.embopress.org/page/journal/14693178/authorguide> for more info on how to prepare your figures.

4) a complete author checklist, which you can download from our author guidelines

(<<https://www.embopress.org/page/journal/14693178/authorguide>>). Please insert information in the checklist that is also reflected in the manuscript. The completed author checklist will also be part of the RPF.

5) Please note that all corresponding authors are required to supply an ORCID ID for their name upon submission of a revised manuscript (<<https://orcid.org/>>). Please find instructions on how to link your ORCID ID to your account in our manuscript tracking system in our Author guidelines

(<<https://www.embopress.org/page/journal/14693178/authorguide#authorshipguidelines>>)

6) We replaced Supplementary Information with Expanded View (EV) Figures and Tables that are collapsible/expandable online. A maximum of 5 EV Figures can be typeset. EV Figures should be cited as 'Figure EV1, Figure EV2' etc... in the text and their respective legends should be included in the main text after the legends of regular figures.

7) Please include a dedicated "Data Availability" section at the end of the Methods (suggested wording: "The [structural coordinates | microarray | mass spectrometry] data from this publication have been deposited to the [name of the database] database [URL] and assigned the identifier [accession | permalink | hashtag]."). Should this not apply, this should still be stated as "This study includes no data deposited in external repositories."

Additional information on source data and instruction on how to label the files are available

<<https://www.embopress.org/page/journal/14693178/authorguide#sourcedata>>

10) Figure legends and data quantification:

- the name of the statistical test used to generate error bars and P values,
- the EXACT p-values,
- the number (n) of independent experiments (please specify technical or biological replicates) underlying each data point,
- the nature of the bars and error bars (s.d., s.e.m.)

- If the data are obtained from n {less than or equal to} 5, show the individual data points in addition to the SD or SEM.

- If the data are obtained from n {less than or equal to} 2, use scatter blots showing the individual data points.

11) Our journal encourages inclusion of *data citations in the reference list* to directly cite datasets that were re-used and obtained from public databases. Data citations in the article text are distinct from normal bibliographical citations and should directly link to the database records from which the data can be accessed. In the main text, data citations are formatted as follows: "Data ref: Smith et al, 2001" or "Data ref: NCBI Sequence Read Archive PRJNA342805, 2017". In the Reference list, data citations must be labeled with "[DATASET]". A data reference must provide the database name, accession number/identifiers and a resolvable link to the landing page from which the data can be accessed at the end of the reference. Further instructions are available at <<https://www.embopress.org/page/journal/14693178/authorguide#referencesformat>>.

12) All Materials and Methods need to be described in the main text using our 'Structured Methods' format. According to this format, the Methods section includes a Reagents and Tools Table (listing key reagents, experimental models, software and relevant equipment and including their sources and relevant identifiers) followed by a Methods and Protocols section describing the methods, ideally using a step-by-step protocol format. The aim is to facilitate adoption of the methodologies across labs. Please download and fill our Reagents and Tools Table template (.docx), which you can find in our author guidelines: <https://www.embopress.org/page/journal/14693178/authorguide#structuredmethods>.

13) As part of the EMBO publication's Transparent Editorial Process, EMBO Reports publishes online a Review Process File to accompany accepted manuscripts. This File will be published in conjunction with your paper and will include the referee reports, your point-by-point response and all pertinent correspondence relating to the manuscript.

Kind regards,

Martina

=====

Referee #1:

The authors have made a very thorough revision of their manuscript. I find that the inclusion of cell divisions in the modelling, and new data (for example Figure 4) in their new manuscript really pushes their work to a new level. I also very much appreciate the efforts made to highlight the truly novel aspects of the work presented in this manuscript. I am very impressed with this version.

Referee #2:

I have included my review in annotated manuscript.

The majority of editorial requests have been addressed by the authors.

Manuscript number: EMBOR-2025-62753V2

Title: Growth history leaves a geometric trace in puzzle cells

Author(s): Nicola Trozzi, Brendan Lane, Alice Perruchoud, Frances Clark, Lukas Hoermayer, Andrea Meraviglia, Tammo Reichgelt, Adrienne Roeder, Dorota Kwiatkowska, Adam Runion, Richard Smith, and Mateusz Majda

Dear Dr. Majda

Thank you for the submission of your revised manuscript to EMBO reports. We have checked all files and all seems fine and I am therefore writing with an 'accept in principle' decision, which means that I will be happy to accept your manuscript for publication once a few minor issues/corrections have been addressed, as follows.

1) We noticed the following author name discrepancy: Adrienne Roeder (in the online system) vs Adrienne H.K. Roeder (in the manuscript). Please resolve this.

2) Please provide callouts for Figures 3G, 6G/J, 7B-C in the text.

3) The resolution of Appendix Fig. S4, S5D, S10, S12 is rather low and should be improved if possible.

4) Appendix Fig. S6, S8, S9, S11: I recommend adding separate scale bars for each panel.

5) Harline K et al is a preprint. Please cite it as follows:

* In the text as (preprint: Harline et al, 2025).

* In the reference list add [PREPRINT] at the end of the reference.

6) During our routine image checks, we noticed that the leaves shown in Figure 4 A-K are the same. I assume that this is intentional as you measure different parameters using the same sample. But to avoid any ambiguities, it might be worth noting this in the figure legend.

Moreover, the leaves shown in Figure 4K appear to be the same shown in Figure 6B (6-8DAS) of Harline K and Adrienne Roeder, *Plant Methods* 2023, PMID 36726130). Please ensure that the publishing license allows you to reuse these data and clearly indicate this reuse in the figure legend.

7) Data availability section: In addition of the DOI we would also need the specific URL that resolves to the Dataset on Zenodo, i.e., https://data.niaid.nih.gov/resources?id=zenodo_15184042.

8) Please address the following points in the figure legends:

* provide the exact p values in the legends of figures 4H, I (unless $p < 0.0001$).

* define the box plots in terms of minima, maxima, centre, bounds of box and whiskers, and percentile in the legends of figures 4G, H, I.

* provide information related to n in the legend of figure 4G.

9) It appears as if the manuscript file with annotations from former referee #2 was not attached to my earlier decision letter. Please find it attached here and please address the two comments the referee added in the text, if applicable, and in a point-by-point response. You find the comments in the PDF file and copied below.

Comment page 4: "I am curious to know if the authors applied random sequence or time of anisotropic/isotropic expansion and whether changing this sequence between attached cells induces some consequences in terms shape? and given this alternation, does it induce consequences for temporal dynamics of auxin/rop signaling that is essential for lobing?"

Comment page 5, related to Suppl. Figure 5D: "it would be better to invoke the temporal aspect of this. From simulations, you show that alternations create lobes. taking this a step further, if cells divide more rapidly, you will have less time for such alternations if I get it right. perhaps mention this?"

Once you have made these minor revisions, please use the following link to submit your corrected manuscript:

Link Not Available

If all remaining corrections have been attended to, you will then receive an official decision letter from the journal accepting your manuscript for publication in the next available issue of EMBO reports. This letter will also include details of the further steps you need to take for the prompt inclusion of your manuscript in our next available issue.

Thank you for your contribution to EMBO reports.

Kind regards,

Comment page 4: "I am curious to know if the authors applied random sequence or time of anisotropic/isotropic expansion and whether changing this sequence between attached cells induces some consequences in terms shape? and given this alternation, does it induce consequences for temporal dynamics of auxin/rop signaling that is essential for lobing?"

We thank the reviewer for raising this point. In the simulations in Figure 1E and 1F, the timing and order of isotropic and anisotropic growth phases were predefined and deterministic. Growth is imposed globally at each simulation step, with a single pair of growth rates in x and y applied to all vertices in the tissue. We did not implement cell-specific or asynchronous switching of growth modes between neighboring cells. Because cells are mechanically coupled in the model, asynchronous switching could in principle alter local stress patterns and thus cell shape, but this is not implemented here. The model also does not include explicit auxin transport or ROP signaling dynamics, so we cannot assess how temporal alternation might couple to these pathways. We added a clarifying sentence in the Results section 'Impact of organ growth dynamics on puzzle cell formation' (lines 132–133).

Comment page 5, related to Suppl. Figure 5D: "it would be better to invoke the temporal aspect of this. From simulations, you show that alternations create lobes. taking this a step further, if cells divide more rapidly, you will have less time for such alternations if I get it right. perhaps mention this?"

We thank the reviewer for this suggestion. We added a sentence to clarify the temporal interpretation: increased division activity shortens the time window for expansion and reduces the opportunity for temporally varying growth anisotropy to shape cell outlines, consistent with the temporal effects observed in our simulations (lines 180–182).

Mateusz Majda
University of Lausanne
Department of Plant Molecular Biology
Switzerland

Dear Dr. Majda,

I am very pleased to accept your manuscript for publication in the next available issue of EMBO reports. Thank you for your contribution to our journal.

You may qualify for financial assistance for your publication charges - either via a Springer Nature fully open access agreement or an EMBO initiative. Check your eligibility: <https://link.springer.com/journal/44319/how-to-publish-with-us>

Yours sincerely,

>>> Please note that it is EMBO Reports policy for the transcript of the editorial process (containing referee reports and your response letter) to be published as an online supplement to each paper. If you do NOT want this, you will need to inform the Editorial Office via email immediately. More information is available here: <https://link.springer.com/partners/embo-press/editorial-policies#Peer%20review>